



# Estimating the risk related to networks: a methodology and an application on a road network

Jürgen Hackl[1], Juan Carlos Lam[1], Magnus Heitzler[2], Bryan T. Adey[1], and Lorenz Hurni[2]

[1]Institute of Construction and Infrastructure Management, ETH Zurich, 8092 Zurich, Switzerland
[2]Institute of Cartography and Geoinformation, ETH Zurich, 8092 Zurich, Switzerland

*Correspondence to:* Jürgen Hackl (hackl@ibi.baug.ethz.ch)

**Abstract.** Networks, such as transportation, water, and power, are critical lifelines to society. Managers plan and execute interventions to guarantee the operational state of their networks under various circumstances, including after the occurrence of (natural) hazard events. Creating an intervention program demands knowing the probable consequences (i.e., risk) of the various hazard events that could occur to be able to mitigate their effects. This paper introduces a methodology to support network managers in the quantification of the risk related to their networks. The method emphasizes the integration of the spatial and temporal attributes of the events that need to be modeled to estimate the risk. This work then demonstrates the usefulness of the methodology through its application to design and implement a risk assessment to estimate the potential impact of flood and mudflow events on a road network located in Switzerland. The example includes the modeling of (i) multiple hazard events, (ii) their physical and functional effects throughout the road network, (iii) the functional interrelationships of the affected objects in the network, (iv) the resulting probable consequences in terms of expected costs of restoration, cost of traffic changes, and duration of network disruption, and (v) the restoration of the network.

## 1 Context

Managers of networks, such as transportation, water, and power, have the continuous task to plan and execute interventions to guarantee the operational state of their networks. This also applies in the aftermath of (natural) hazard events. Since the resources available to managers to protect their networks are limited, it is essential for managers to be aware of the probable consequences (i.e., risk) in order to set priorities and be resource-efficient (Eidsvig et al., 2016). Conducting a risk assessment can help identify the probable hazard events, and evaluate their impact on networks and their users.

Nonetheless, conducting a risk assessment can be a particularly challenging task due to the large number of *scenarios* (i.e., chains of interrelated events) that need to be taken into account, and their associated probabilities and estimated consequences. Furthermore, multiple types of hazards need to be considered (Komendantova et al., 2014; Mignan et al., 2014; Gallina et al., 2016), along with the complex nature of networks, specifically, their large number of objects, their spatial distribution, and functional interrelationships. Moreover, there are additional spatial and temporal characteristics that need to be considered in building these scenarios, along with methods to model the cascade of events, the network interdependencies, and the propagation of uncertainties (Hackl et al., 2015).



The estimation of the probable consequences adds another layer of complexity. Consequences are often expressed in monetary values, and these are distinguished between direct costs (e.g., costs related to cleanup, repairs, rehabilitation and reconstruction) and indirect costs (e.g., in the transport sector, costs related to additional travel time, vehicle operation and an increase in the number of accidents; in the drinking water sector, costs related to delivery of clean water and purification tablets and

an increase in the number of cases of waterborne diseases). Since indirect costs have a wide spatial and temporal scale (Merz et al., 2010), and are potentially larger than direct costs (Vespignani, 2010), it is important to also model how the restoration of the damaged network occurs (Lam and Adey, 2016). Examples of restoration modeling include the work of He and Liu (2012); Bocchini and Frangopol (2012); Vugrin et al. (2014); Sun (2017).

As a result of all of these challenges, risk assessment methods for networks have been the subject of increasing research

interest in recent years. Much of this research has focused on road networks (please refer to the work of Opdyke et al. (2017) for trends in the transport sector) and the probability that the network will provide an adequate level of service. Examples of works in this domain include those of Vugrin et al. (2014); Jenelius and Mattsson (2015); Lam and Adey (2016); Hackl and Adey (2018).

Mattsson and Jenelius (2015) refer to two different types of risk assessments for transport networks. One type has its roots

in graph theory, and is focused on studying the structural (topological) properties of the networks. This analytical approach only requires network topology data, and considers the importance of different edges (Jenelius et al., 2006; Rupi et al., 2015), cascading failures (Dueñas-Osorio and Vemuru, 2009; Hackl and Adey, 2017), and interdependencies between different networks (Thacker et al., 2017). The other type of risk assessment attempts to understand the dynamic behavior of the networks through the use of transportation system models. For the latter approach, extensive data is needed and the computational efforts

are larger; however, this approach provides a more complete description of the society-related consequences (Mattsson and Jenelius, 2015).

## 2   Contribution

This paper introduces the risk assessment methodology proposed by Hackl et al. (2016), which was based on the work of Adey et al. (2016) and Hackl et al. (2015). The methodology can be used to investigate multiple scenarios, starting from the modeling

of a source event to (e.g., rainfall, fault rupture), and ending in the estimation of the probable consequences. The methodology is then followed to estimate the risk related to a road network located in the Rhine Vally area, specifically in and around the city of Chur, Switzerland, as a result of probable flood and mudflow events of various return periods, which were simulated to be caused by rainfall events. All the challenges presented in Section 1 were accounted for in this example, with the exception of network interdependencies (e.g., interaction between road and rail networks, or between road and water networks). This is

subject of future work.

The remainder of this work is structured as follows. A brief overview of the risk assessment methodology is presented next in Section 3. Section 4 contains a description of the main features of the example used to illustrate the application of the





methodology, along with the results obtained. In Section 5, discussions about the methodology and the findings of the example are given, followed by a summary of the work and suggestions for future research in Section 6.

## 3 Risk assessment methodology

### 3.1 Overview

The methodology is described in detail in Hackl et al. (2016). The purpose of the developed process is to support network managers in the quantification and subsequent management of risk. The method is founded on the principles of systems engineering (Adey et al., 2016):

- *following a basic structured process to solve problems* (e.g., best solution is implemented after carefully generating, analyzing and evaluating possible solutions, and considering the problem at hand, goals and constraints),

- *working in phases* (e.g., first, qualitative analysis over a short period of time, and later if required, quantitative analysis over a longer period of time),

- *working from a high level to a low level of abstraction* (e.g., first analysis delivers less detailed information, and later analysis delivers more detailed information), and

- *thinking of possibilities* (e.g., there are many ways to perform an assessment).

Therefore, the process is structured keeping in mind that (i) different decisions will require different types of models, (ii) models provide different levels of detail, and (iii) this is an iterative process, requiring reworking as data and model insufficiencies are discovered, and new data and models become available.

Considering these principles, the risk assessment methodology is structured in six major tasks, one of which, *define system representation*, contains five sub-tasks (please see Figure 1). These tasks and sub-tasks are succinctly described next.

### 3.2 Description of tasks

#### 3.2.1 Set up risk assessment

This task is used to determine what needs to be analyzed to make a statement on whether risk reducing interventions are needed, or not. Conducting this task has an effect on the scope and the level of detail of the assessment, the definition of the system representation, and the requirements to perform the assessment in terms of input and output. The task results in a rough

outline of the planned risk assessment.

#### 3.2.2 Determine approach

This task has the purpose to enhance the effectiveness and efficiency of the risk assessment by enabling stakeholders to make decisions about how the risk assessment will be conducted. Hence, stakeholders with substantial experience on risk assessments can be beneficial to the outcome of this task, which ends with an agreed approach on how to perform the risk assessment, who

will be involved and what is to be done.




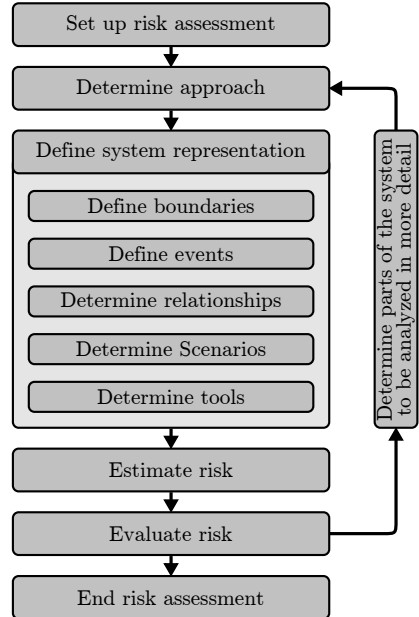

**Figure 1.** Schematic overview of the risk assessment methodology.

### 3.2.3 Define system representation

This representation is a model of the relevant realizations of the stochastic scenarios within the investigated time period. Modeling the system representation is necessary, and requires the consideration of the spatial and temporal correlation between events. This tasks includes agreements on system behavior, drawing fixed system boundaries, and documenting assumptions and limitations.

*Define boundaries*

This sub-task involves defining the parts of the system that are going to be analyzed, both spatially and temporally, and how these parts are to be subdivided. This sub-task helps (i) specify where objects are located, where source and hazard events can occur, and where consequences could take place, and (ii) define the time periods over which the system representation is to be analyzed.

*Define events*

This sub-task involves identification of all the events that are to be analyzed, which can be categorized as *source*, *hazard*, *infrastructure object*, *network* and *societal events* (please see Table 1). All events are described in space and time, and measures of the intensities should be assigned to describe event attributes of interest. These measures depend on the problem investigated and the level of detail of the analysis.

*Define relationships*



| Event | Description | Example |
|---|---|---|
| Source | An event that may lead to a hazard event. | Rainfall, tectonic plate movement |
| Hazard | An event that may lead to an infrastructure object event, and some times, to another hazard event (e.g., earthquake triggers landslides). | Flood, earthquake, landslide |
| Infrastructure object | An event that is a change in the infrastructure object that may lead to a change in network use or a change in human behavior. | Pier scour, pavement inundation, mudblocking |
| Network | An event that is a change in how the network can be used that may lead to a change in human behavior. | Road closures, reduced speed limits |
| Societal | An event that is a change in human behavior (to which a value can be placed). | direct and indirect costs |

**Table 1.** Basic event types.

Models representing the relationships between the events help build the scenarios, and estimate their probabilities of occurrence. This sub-task is focused on (i) determining a target level of relationship accuracy, and (ii) developing these relationships using available data. Extra effort should be spent on describing these relationships in more detail when approximations are not adequate as exposed by the obtained results.

*Define scenarios*

The definitions of events and relationships provide the basis for constructing scenarios. The identification of the scenarios to be analyzed as part of the risk assessment should be done with care (e.g., without explicit estimation of their probability of occurrence or a consequence value beforehand to avoid a biased selection), knowing that excluding specific scenarios may result in an incorrect estimation of risk.

*Determine tools*

Once the scenarios to be analyzed are determined, the specific tools, models and approaches to be used to estimate the risk are identified. This includes the selection of the software packages to be used if computer support is required, or a decision to develop software tailored to the specific needs of the risk assessment. This task ends with the selection of tools, models, and software.

**3.2.4   Estimate risk**

In this task, the probabilities of scenarios and their consequences are estimated, and aggregated when desired. This task can be undertaken with varying degrees of detail and with or without computer support, depending on the problem, and the data, information, and resources available. This task ends with the estimation of the risk, their uncertainty and their sensitivity to modeling assumptions.



### 3.2.5 Evaluate risk

This task supports the interpretation of the risk estimates by involved stakeholders—process that may be affected by risk perception and valuation. Although systems are never modeled perfectly, this task helps bringing network managers closer to an optimal solution to the original problem that would be acceptable to all stakeholders. This task can result in three decisions:

1. assessment was conducted satisfactorily, and no risk reducing interventions are needed based on the results,

2. assessment was conducted satisfactorily, and risk reducing interventions are needed based on the results, or

3. assessment was not conducted satisfactorily, and therefore, more analysis is required.

### 3.2.6 Determine parts of the system to be analyzed in more detail

If the risk evaluation shows that the assessment could not be carried out satisfactorily, changes in the approach or system representation may have to be made (e.g., parts that are likely to generate the most reduction in the uncertainty of risk estimates). The parts to be analyzed in more detail can be determined taking into consideration the available resources.

## 4 Application

### 4.1 Context

With a changing climate exacerbated by an increase in urbanization, the frequency of extreme climate-related hazard events, such as floods and rainfall-triggered landslides, is expected to rise, impacting economic corridors, disrupting supply chain, and stressing emergency and rescue operations, among other effects. As a result, special focus is now given to this type of hazard events and their associated risks.

Floods are the leading hazard events in Europe in terms of economic losses (EEA, 2010). In times of scarce public resources and increasing incidences and damages caused by floods (Bowering et al., 2014), public policy makers and network managers have the need to become increasingly aware of their causes and consequences so that they can appropriately manage their risks, for example, through the adaptation of networks (Elsawah et al., 2014) or the planning of actions following a hazard event (Taubenböck et al., 2013). A number of methodologies have been developed to quantify the damages and costs due to flooding (Merz et al., 2010; Hammond et al., 2015). Nevertheless, most of these studies focus on building assets and direct costs (e.g., Rogelis et al. (2014)).

In the transport sector, some work has been done to consider probable direct and indirect costs (e.g., Scawthorn et al. (2006); Deckers et al. (2009); Bowering et al. (2014)), but these works have often neglected the spatial and temporal attributes of these networks. Only a few scholars have investigated flood risk in combination with the actual dynamic behavior of networks. Among them, Dawson et al. (2011) implemented an agent-based model to simulate human response to flooding considering different storm surge conditions. Furthermore, Suarez et al. (2005) studied the impacts of flooding and climate change on the urban transport system of the Boston Metro Area, using a conventional analytical framework for simulating traffic flows under different flooding scenarios, changes in land use, and demographic and climatic conditions.





Floods are the first most common rainfall-triggered hazard events in mountainous areas. Landslides are in second place. Since road networks in these areas generally have a low level of connectivity, such as in the case of remote villages that are only connected by few mountain roads, there is a high probability that connections between some areas within a road network can be completely interrupted when one (or more) elements of the network fails. (Rupi et al., 2015). Usually, risk estimates

related to networks, including road networks, due to landslides are obtained by overlapping hazard and consequence maps (Ferlisi et al., 2012; Pellicani et al., 2017), and therefore, the indirect costs for specific hazard scenarios are not considered.

## 4.2 Contribution

The growing interest in investigating the effects of climate-related events makes the methodology proposed here relevant to network managers because the structured tasks can be followed to support capturing such complex scenarios and estimate their

consequences.

The application presented in this section is used to demonstrate the usefulness of the methodology considering a specific problem. The application, which is an improved version of the example presented by Hackl et al. (2016), shows the design and implementation of an assessment focused on estimating the risk related to a road network in the Canton of Grisons in Switzerland. In the study, the network was exposed to rainfall events, which caused multiple hazards, specifically riverine

flood and mudflow events. At the same time, these events led to direct costs linked to clean-up, repair, rehabilitation and reconstruction activities, and indirect costs associated with loss of connectivity and temporal prolongation of network user travel, linking the modeling of these latter effects with the dynamics of the network. As specified later, a large number of uncertain rainfall events of multiple return periods was considered in the analysis. While the application was focused on a regional road network affected by floods and mudflows, the methodology is applicable to different types of hazards, different

types of networks, different types and sizes of regions, and different levels of abstraction.

The remainder of this section is structured accordingly to the tasks of the risk assessment methodology in Section 3. First an overview of the the investigated network is given in Section 4.3, followed by the assessment setup, determined approach and the chosen system representation. A major focus of the application is on the models of events and relationships. Therefore, in Section 4.7, the models implemented are discussed in general terms, while the more detailed mathematical representation is

given in Appendix B. Finally, the definition of risk and the actual risk estimates are given in Sections 4.8 and 4.9, respectively.

It is worth noting that the data used for this example were representative of actual entities and processes in the region, or were derived from such data. Moreover, the models selected for the example were chosen considering the need to keep the computational time low. The authors of this paper are fully aware that more sophisticated and precise models may be available in the literature, some of which demand much more detailed data than the data that were available. Nonetheless, the selection of

data and models was deemed sufficient for illustrating the use of the methodology, which is the main objective of the example. In order to draw some conclusions about the risk in the region, formal data sources and the official engagement of the network manager and other relevant stakeholders would be needed. While the risk assessment methodology remains unaffected by these limitations, the computational support designed and implemented for the example, was constructed in modules, allowing for the updating of data and models as needed.





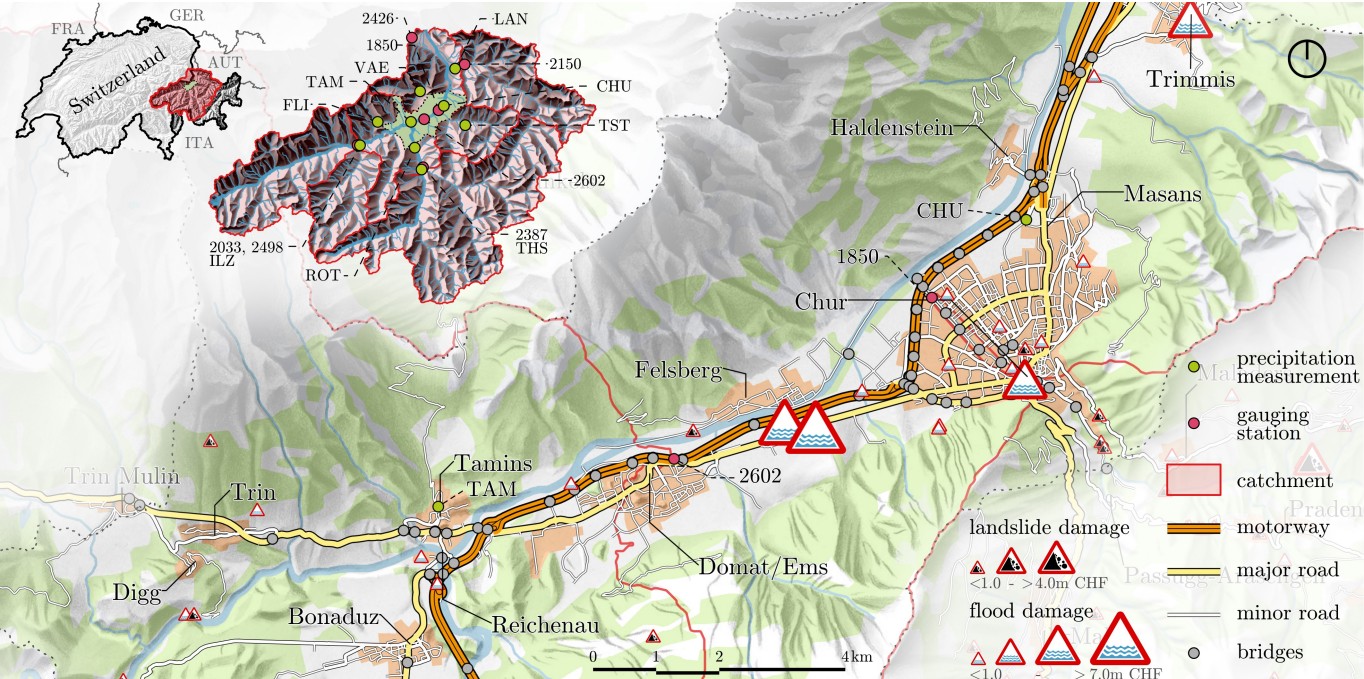

**Figure 2.** Area of study.

## 4.3 Area of study

The investigated road network is located in the Rhine Valley between Trin and Trimmis (Figure 2). This target area is located around the city of Chur, the capital of Grisons, the largest and easternmost canton of Switzerland. It is the largest city in Grisons with approximately 34,500 inhabitants, and a large business center. Chur is also an important transportation hub,

linking Switzerland, Germany, Austria, and Italy. The Canton of Grisons is crossed in a north-south direction by the A13 motorway. The considered road network comprises circa 121 bridges and 605 km of roads, including 51 km national roads.

Lake Toma in Grisons is generally regarded as the source of the Rhine. Its outflow is called Rein da Tuma, and after a few kilometers, the outflow forms the Anterior Rhine. The Anterior Rhine is about 76 km and has a catchment area of $1,512\,\mathrm{km}^2$. The Posterior Rhine is the second tributary of the Rhine, with less length, but a larger discharge volume than that of the Anterior

Rhine. The basin of the Posterior Rhine is approximately $1,698\,\mathrm{km}^2$. The river begins to be called Rhine at the confluence of the Anterior Rhine and Posterior Rhine in Reichenau.

For both rivers, detailed runoff and flood information were available from the stream gauge stations in Ilanz (No. 2033, 2498) and Fürstenau (No. 2387). Both towns are at the considered boundaries of the investigated area. Another stream gauge station is located in the study area at Domat/Ems (No. 2602), which was used as a reference point for the hydraulic model.

Nine precipitation gauge stations with sufficient extensive historical series were available in the basins under consideration. Precipitation data since 1887 were available to determine and calibrate extreme rainfall events.



The investigated area is exposed to flooding and landslides on an annual basis as the historical records of the Swiss Flood and Landslide Damage Database[1] shows. Figure 2 illustrates the recorded hazards from 1975 to 2013, which have occurred in the area and affected networks[2].

The database contains data on 45 hazard events that have impacted networks located within the region of interest. From these, 27 events fall into the category of floods/debris, 13 events fall into the category of landslides, and the remaining 5 events are classified as rockfalls. The costs are in the range between CHF 10 thousand and CHF 6.85 million. According to the dataset, 4 events have caused "high/catastrophic" damages while 5 and 36 have caused "medium" and "minor" damages, respectively[3]. Consequences include 12, 31 and 2 disruptions on the railway network, the road network and traffic, and the power supply, respectively. Over the period of 38 years, 2 bridges were severely damaged and 3 bridges collapsed. One of the worst floods in recent history occurred in 2005. This flood was responsible for approximately CHF 695 million in damages to roads and railways in Switzerland(Bezzola and Hegg, 2007).

## 4.4 Set up risk assessment

The purpose of the assessment itself was to quantify the risk of a complete chain of events over space and time, from source events to their consequences, considering: rainfall, runoff, flooding, mudflows, physical damages and functional losses to bridges and roads (road galleries, tunnels and other structures were not of primary concern), traffic changes and restoration works. Consequences were monetized into direct and indirect costs, which were associated with restoration interventions, prolongation of travel, and lost trips.

The assessment was done considering only source events that could happen within a one year period (i.e., the interest was on understanding what could occur in a given year before changes in the system representation due to renewal works, network extension, and urban sprawl, among other factors). Moreover, it was assumed that only one source event leading to hazard events could happen throughout this period (i.e., a sequence of events was not explored). The source events would have a maximum return period of 10,000 years.

## 4.5 Determine approach

A quantitative approach was used to explicitly take into consideration the spatio-temporal properties of the events. For each element in the chain of events, spatio-temporal models were applied, representing the meteorological, hydrological, hydraulic, geotechnical, structural, functional, and socioeconomic processes.

The quantitative approach was supported by the use of computers, specifically a simulation engine, which integrated all required models, utilities to access/write any input/output data set and parameters, as well as modules to define how the models were to be coupled and executed (Heitzler et al., 2018). This engine supported running many simulations of what might happen.

---

[1] WSL Unwetterschadens-Datenbank der Schweiz

[2] Events that caused damages to buildings or forestry/agricultural are not considered.

[3] A definition of these terms can be found in Hilker et al. (2009).



Since an exhaustive evaluation of all possible scenarios has not been possible to date for complex system representations (as later described), this work aimed to develop a simulation engine that could support running a large number of scenarios.

Such a simulation-based framework could then account for aleatory uncertainty, due to natural and stochastic variability, and epistemic uncertainty due to incomplete knowledge of the system. A quantitative risk assessment should take into ac-
count all relevant scenarios, their associated probabilities and possible consequences as well as a thorough investigation of the uncertainties associated with the resulting estimations.

## 4.6  Define system representation

### 4.6.1  Define boundaries

*Spatial boundaries*

The system is spatially bounded by several event-related boundaries. To model the spatio-temporal behavior of rainfall and the changes in the river discharge values, a larger area than the area used for assessing the hazard events was considered. The hazard and infrastructure object events were bounded to the corresponding catchment area of the region, as illustrated in Figure 2. The boundaries for the network and societal events exceeded the boundary of the infrastructure object events to proper model the effects at the network level. The costs were only accounted for the Chur region. Impacts at the national level
were not considered (e.g., blockade of the transit route from northern Europe to the south).

*Temporal boundaries*

Scenarios were divided into time steps, with the duration of each time step varying, depending on the occurred events in the scenario. Source and hazard events were observed every hour, in order to account for gradual changes of the system (e.g., the traveling paths of rainfall, the increase of river stage). Infrastructure object, network, and societal events were evaluated on an
hourly basis (during the hazard events) and on a daily basis (after the hazard events) since the restoration time was expressed in working days.

### 4.6.2  Define events

The selected events were the following:

- source event: *rainfall*,
- hazard events: *floods* and *mudflows*,
- infrastructure object events: *bridge local scour*, *pavement inundation*, and *pavement mudblocking*,
- network events: *time-varying network-level functional losses*, and
- societal events: *restoration works*, *changes in traveling routes*, and *loss of connectivity*.





### 4.6.3 Define relationships and scenarios

Once these events were defined, scenarios were built considering the relationships among events.

- rainfall events leading to flood events, supported by *runoff* calculations,
- rainfall events leading to mudflow events, triggered by the exceedance of *rainfall intensity-duration thresholds*,
- calculated spatially and temporally distributed discharge, inundation depth and mudflow volume values leading to object-level physical damages (i.e., bridge local scour, pavement inundation, pavement mudblocking) and object-level functional losses (i.e., lane closure, speed reduction), using *fragility functions* and *functional capacity loss functions*
- object-level functional losses leading to network functional loss, considering the *topology* of the network,
- object-level physical damages and object-level functional losses leading to restoration works, considering specific *restoration criteria*, and
- network functional losses leading to changes in traffic flow, supported by an *origin-destination matrix*.

### 4.6.4 Determine tools

Simulation-based risk assessments require the coupling of multiple heterogeneous models, where a given model encapsulates the behavior and state of a part of the system. During execution, these models exchange data allowing to simulate the chain of 15 events through all represented system scenarios.

To support this exchange, models were grouped into modules. Each module comprised distinct execution instructions, and required certain input data to be properly executed. Such input data may consist of parameters for the models, static input data (e.g., location of mudflows, extent of the road network) or even states of models originating from other modules (e.g., to determine the damaged objects, the water extent from a flood model is needed) as illustrated by Heitzler et al. (2018).

The models and modules used were programmed in Python. Since most network managers use geographic information systems (GIS), a GIS data interface was developed to facilitate the import and export of data. Furthermore, the program code was optimized for massively parallel computing in order to reduce the computational time of the optimization process (i.e., each simulation ran on a designated CPU, and therefore, an increase in the amount of CPUs, increases the number of simulations for a given timeframe)

With this approach, a chain of events could be assembled. Further details about the simulation engine and their implementation are provided by Heitzler et al. (2018).

### 4.7 Models

### 4.7.1 Rainfall model

A representations of rainfall that considered spatio-temporal variability and had a fine resolution was desired. Therefore, the 30 experimental precipitation catalog described by Wüest et al. (2010) was used as an input for the rainfall model. The data set covers all of Switzerland for the period between May 2003 and May 2010. This catalog was derived from a combination of




precipitation-gauge-based high-resolution interpolations and an hourly composite of radar measurements. The aggregation of the two data sources allowed for a catalog of very high resolution at the spatial scale ($1\,\mathrm{km}^2$) and at the temporal scale ($1\,\mathrm{hour}$).

The initial precipitation fields were sampled from this catalog using randomized start hours and durations. The precipitation values for each raster cell for each time step were scaled to associate the return period of a generated discharge value at a
location of interest with that of the rainfall event (Hackl et al., 2017). The result of this process was a scaled time series of precipitation fields (i.e., one raster file for every time step).

### 4.7.2 Runoff model

To account for the spatially and temporally varied rainfall events, the implemented runoff model was based on the modified Clark (ModClark) model by Kull and Feldman (1998). After parsing a watershed into a uniform grid, this model uses a
linear quasi-distributed transformation method to estimate the runoff based on the Clark conceptual unit hydrograph (Clark, 1945). The method accounts for spatial differences in precipitation and losses (Paudel et al., 2009), allowing to model runoff translation and storage. The implementation of this model was calibrated by comparing the calculated values with measured values from the stream and precipitation gauge stations in the area.

### 4.7.3 Flood model

The estimated runoffs at the watershed outlets were added to the base flows of the rivers to produce the final hydrographs for each river section, which were entered into a one-dimensional hydraulic model for gradually varied steady flow in an open channel network. The model was used to simulate the floodplain inundation by first computing the water surface profile of a given cross-section to the next, and then interpolating inundation values in between cross-sections to obtain the inundation field for the area of study. The cross-sections were obtained from a digital elevation model (DEM) of the area. In total, 198
cross-sections with an average distance of $150\,\mathrm{m}$ were considered.

The boundary conditions were represented by (i) the discharges at Ilanz and Fürstenau, (ii) the water level at the Rhine outlet, and (iii) the evolution of the additional discharges per cross-section from the computed hydrographs. The (Manning) roughness coefficients were estimated based on soil cover and land use. Finally, the hydraulic model was calibrated based on historical records of the stream gauge stations in the area. The outputs of interests were raster files representing the water depths at each
cross-section of the rivers, and inundation fields for all the time steps of the hazard event simulation.

### 4.7.4 Mudflow model

The model consisted of three main processes: (i) determination of potential mudflow locations, (ii) modeling the potential geometries and volumes, and (iii) estimation of the probability to trigger an event.

Potential locations and geometries were obtained from Losey and Wehrli (2013). Potential locations were determined using
geological data and relief parameters as described in Giamboni et al. (2008), and geometries were calculated using the random walk routine of Gamma (2000). The volume of each mudflow event was then estimated by calculating the runout length of



the fan using the empirical relation of Rickenmann (1999). The increase in elevation per cell was calculated by dividing the mudflow volume by the area of the fan.

The probability of occurrence was estimated by first using the empirical intensity-duration function for sub-alpine regions proposed by Zimmermann et al. (1997) to determine which events could be triggered at a given time step. When intensity-

duration thresholds were exceeded at the site of a potential mudflow, a probability of being triggered was assigned to the event. This probability was related to the corresponding estimated slope factor of safety (FS) (Skempton and Delory, 1952).

In total, 54 potential mudflows were considered in the target area of study. The output of the model was a time series of raster files, each of which contained cell values corresponding to mudflow elevations.

### 4.7.5 Bridge local scour model

Assuming a negligible contraction of the river bed cross-sections (Gehl and D'Ayala, 2015), only local (pier) scour was considered for the example, resulting in the selection of five bridges that could fail as a result of this phenomenon. To qualify different levels of local scour, four damage (limit) states were defined as described in Table 2.

| State | Label | Description |
|-------|-------|-------------|
| 0 | *operational* | no changes in bridge response |
| 1 | *monitored* | first noticeable changes in bridge response |
| 2 | *capacity-reduced* | significant changes in the bridge response |
| 3 | *closed* | lack of pier stability to support the bridge |

**Table 2.** Damage states for bridge local scour.

Log-normal fragility functions were used to relate these damage states with a range of possible discharge values for two types of bridges, namely bridges with one pier and those with two piers. The methodology used the local scour model of Arneson et al.

(2012) and a Monte Carlo scheme to generate 100,000 uncertain local scour depths. These depths were compared with depth thresholds assumed to correspond to each damage state, leading to the estimation of damage state exceedance probabilities. The derived fragility parameters for the scour damage states are given in Table B2. The fragility functions are illustrated in Figure 3.

### 4.7.6 Pavement inundation model

To account for the spatially distributed representation of the hazard events (e.g., mudflow and inundation intensity measure fields), the road network was split up into a set of 74,466 unidirectional road sections $n$ (excluding bridges), aligned with the raster cells of the hazard events. The impact of each hazard could then be assessed for each road section individually.

Having said this, determining the extend of physical damage of a road section due to a flood event is unfortunately an under researched area, and hence, it is rather challenging to describe such relationship in a numerical form with the little information

available to date. Some works (e.g., De Bruijin (2005); Kok et al. (2005); Koks et al. (2012); Tariq et al. (2014); Li et al. (2016)) sought to numerically describe the relationship between damages and inundation depth; however, there are observed



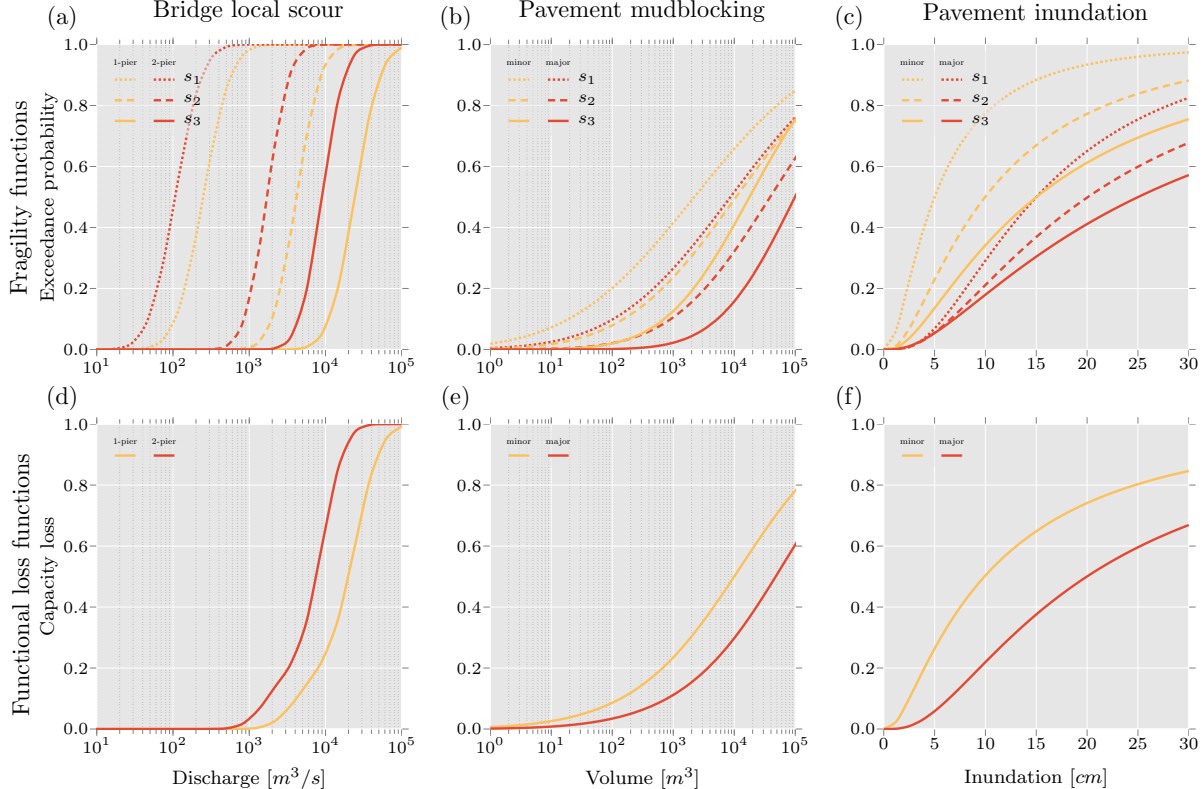

**Figure 3.** Fragility functions and functional capacity loss functions.

differences in the methodologies and results (e.g., some works bundled direct and indirect consequences in their estimates), leading to the conclusion that these numerical relationships can only be applied to specific contexts and confined geographical areas.

For this example, a different approach was taken than those presented above. The approach involved proposing log-normal
5   fragility functions based on previous works seeking to qualitatively illustrate the impact of flood events on road sections (e.g, ALA (2005); ADEPT (2011); Vennapusa et al. (2013); Walsh (2011)). Despite the use of these studies, the proposed functions remain to be coarse illustrations that cannot be used in practice without further analysis. To qualify different levels of damage, four damage states were defined as described in Table 3. The derived fragility function parameters are given in Table B3. The fragility functions are displayed in Figure 3.

10  **4.7.7   Pavement mudblocking model**

To determine the level of pavement mudblocking, a distinction was made between high-speed (major) and local (minor) roads. The damage states defined by this work were based on the descriptions of Winter et al. (2013). These states are shown in Table 4.





| State | Label | Description |
|---|---|---|
| 0 | *operational* | no observed damages, negligible sign of sediments |
| 1 | *monitored* | presence of sediments and debris |
| 2 | *capacity-reduced* | elements of the road section slightly damaged |
| 3 | *closed* | loss of subgrade layer (given that road section was inundated over one day and traffic was observed while inundated) |

**Table 3.** Damage states for pavement inundation.

| State | Label | Description |
|---|---|---|
| 0 | *operational* | no observed damages |
| 1 | *monitored* | encroachment limited to verge/hard strip |
| 2 | *capacity-reduced* | blockage of hardstrip and one running lane |
| 3 | *closed* | complete blockage of carriageway and/or repairable damage to surfacing |

**Table 4.** Damage states for pavement mudblocking (using the descriptions of Winter et al. (2013)).

Log-normal fragility functions were estimated using expert data from the survey conducted by Winter et al. (2013), where experts were asked to relate debris flow volumes with damage state exceedance probabilities for different damage states and road categories. The generated fragility function parameters are given in Table B4. The fragility functions are shown in Figure 3.

### 4.7.8 Functional loss model for bridge local scour and pavement mudblocking

Functional losses related to bridge local scour and pavement mudblocking during the hazard event period were estimated based on novel functional capacity loss functions initially proposed by Lam and Adey (2016) and Lam et al. (2018). Expected functional capacity losses were determined as functions of time-dependent hazard intensities $\Xi_t$, derived damage state $ds$ probabilities obtained from respective fragility functions, and functional capacity loss values $\lambda_n$ associated with the investigated damage states (please see Table 5).

$$\langle\lambda\rangle_{n,t} = \mathbb{E}\left(\lambda_n|\Xi_t\right) = \sum_{ds} \mathbb{E}\left(\lambda_n|ds\right) \cdot \mathbb{P}[ds|\Xi_t] \tag{1}$$

where $\langle\lambda\rangle_{n,t} \in [0,1]$ is the expected functional capacity loss of section $n$ at time $t$ given that the section.

| State | Label | bridge | | road | |
|---|---|---|---|---|---|
| | | **1-pier** | **2-pier** | **major** | **minor** |
| 0 | *operational* | 0.0 | 0.0 | 0.0 | 0.0 |
| 1 | *monitored* | 0.0 | 0.0 | 0.3 | 0.3 |
| 2 | *capacity-reduced* | 0.2 | 0.2 | 0.5 | 0.5 |
| 3 | *closed* | 1.0 | 1.0 | 1.0 | 1.0 |

**Table 5.** Functional capacity losses.





The values in Table 5 were either directly obtained or inferred from a survey conducted by D'Ayala and Gehl (2015). In future analyses, these values will need to be re-estimated by local experts and/or analytical methods. The functional capacity loss functions are illustrated in Figure 3.

### 4.7.9 Functional loss model for pavement inundation

During the hazard event period, the relationship between inundation depths and feasible speed of vehicles on the road was derived from the data presented by Pregnolato et al. (2017). An exponential function was fitted to these data to describe the limit vehicle speed in a road as a function of inundation depth.

$$
v_{n,t}^T = \begin{cases} v^{\max} \cdot e^{-0.10814 \cdot i_{\boldsymbol{c}_n,t}} & \text{for} \quad i_{\boldsymbol{c}_n,t} \le 30\,\text{cm} \\ 0 & \text{otherwise} \end{cases}
\tag{2}
$$

where $v_{n,t}^T$ is the maximum acceptable velocity that ensures safe control of a vehicle through section $n$ at time $t$ when con-
sidering the inundation depth $i_{\boldsymbol{c}_n,t}$. For this example, the maximum allowed speed on any road $v^{\max}$ was set to be $120\,\text{km/h}$. Additionally, functional capacity loss functions were developed for inundated pavements following the same process described in Section 4.7.8 for bridge local scour and pavement mudblocking, and using the values of Table 5 corresponding to major and minor roads. At the end of the hazard event period, inundated pavements were assigned such functions to determine the need to intervene on then, their probable restoration costs, times and functional losses during the restoration period. The functional
capacity loss functions are shown in Figure 3.

### 4.7.10 Functional loss model for the network

After functional losses were computer for all individual sections $n$ of the infrastructure objects $e \in \mathcal{E}$, these were aggregated at the network level. These involved three steps: (i) representing the collective of individual bridges and road sections as a network graph, (ii) determining the functional loss changes at the object level throughout the simulation period, and (iii)
combining these functional losses considering the network representation to determine the functional losses at the network level.

The road network was modeled as a graph composed of 1,520 vertices (i.e., 37 centroids, 1,056 junctions, and 427 changes in road geometric features) and 3,202 directed edges $e$, also referred to as links or infrastructure objects. Because the road network is located in a mountainous area, the topology of the network is such that certain areas are served by a single road,
which means that, if part of this road is disrupted, there is no valid rerouting alternative and part of the demand remains unsatisfied. This results in lost trips. For this network, in particular, these (cut) links represented about $11\,\%$ of the entire edge set.

The algorithm used to compute the changes in functional losses at the object level is presented in detail by Lam et al. (2018). Changes occurred during the hazard event (due to time-varying intensity measures) and the restoration period (due



to restoration-related activities). Furthermore, details about the aggregation of these losses for the network are presented by Heitzler et al. (2018).

### 4.7.11 Traffic model

Vehicle travelers were assumed to behave according to the user equilibrium principle, which states that they choose a route from their origin to their destination that minimizes their travel cost. This means that the travel cost between each origin-destination pair is uniquely defined (Jenelius et al., 2006). This cost depends on the travel costs of all edges in an origin-destination path, which change with traffic flow. A stable state is reached only when no traveler can reduce his/her own costs of travel by unilaterally changing routes (Sheffi, 1985). The travel time, estimated using the formulation proposed by the Bureau of Public Roads (1964), was multiplied by the value of travel to obtain the travel time costs.

Regarding the demand model, the origin-destination matrix was estimated by a gravity model based on population density, and was related to the number of vehicles on an average hourly and an average daily basis. This matrix was calibrated and updated using information of recent traffic counts along a set of edges. The study area was divided into 37 zones. Every internal centroid corresponded on average to an area of $15 \, \text{km}^2$ with a population of about $1,000$.

To reduce computational complexity, only changes in route choices were considered within the risk assessment (i.e., travel demand was assumed to be inelastic). This was deemed to be a reasonable assumption, since a large majority of the travel demand was caused by trips to work, which are made under normal circumstances. Therefore, changes in destination choices, mode choices or trip frequencies were not considered. Moreover, it was assumed that the duration of network closures was long enough for all travelers to be aware of them so that a new user equilibrium could be reached.

### 4.7.12 Restoration model

Once the discharge values along the river return to normal conditions and no further damages to objects could occur, the impaired network was restored. This procedure was simulated by a basic restoration model, which was founded in the work of Lam and Adey (2016). In contrast to the hazard occurrence period, the temporal resolution during restoration was set to one day.

Expected functional capacity losses during the restoration period were estimated based on the values presented in Table 6.

| State | Label | bridge | | road | |
|---|---|---|---|---|---|
| | | 1-pier | 2-pier | major | minor |
| 0 | *operational* | 0.0 | 0.0 | 0.0 | 0.0 |
| 1 | *monitored* | 1.0 | 1.0 | 0.5 | 0.5 |
| 2 | *capacity-reduced* | 1.0 | 1.0 | 0.5 | 0.5 |
| 3 | *closed* | 1.0 | 1.0 | 1.0 | 1.0 |

**Table 6.** Functional capacity losses during restoration.





The prioritization of restoration activities was done by first restoring objects that caused a loss of connectivity, and second restoring objects based on on their average traffic volume (i.e., the average daily traffic volume for each object under normal conditions). It was assumed that 10 reconstruction crews could work simultaneously and each crew could at most be assigned to one object. For each day until the restoration was finished, the model updated the objects' states, and executed the traffic
model described in Section 4.7.11. The restoration process ended once the network was restored.

## 4.8  Estimate risk

### 4.8.1  Definition of risk

Risk is here expressed in terms of expected monetized losses, calculated as a product of the probability of occurrence of a certain scenario and the associated costs of those scenarios, for those scenarios –both the direct ($c^{\mathrm{dc}}$) and the indirect costs ($c^{\mathrm{ic}}$)
should be considered. Their cost functions $C^{f,\mathrm{dc}}$ and $C^{f,\mathrm{ic}}$ are associated with the modeled societal events $E^{\mathrm{soc}}$ that occur as a result of infrastructure object events $E^{\mathrm{inf}}$ and network events $E^{\mathrm{net}}$, and can be traced back to hazard events $E^{\mathrm{haz}}$ and source events $E^{\mathrm{src}}$. This representation can also be used when considering multiple hazard events.

$$\mathcal{R} = \int\limits_{E^{\mathrm{haz}}} \mathbb{P}[E^{\mathrm{haz}}|E^{\mathrm{src}}] \cdot (C^{f,\mathrm{dc}}(E^{\mathrm{soc}}|E^{\mathrm{inf}}) + C^{f,\mathrm{ic}}(E^{\mathrm{soc}}|E^{\mathrm{net}})) \tag{3}$$

Direct costs are the intervention costs for the affected objects, which are required to return those objects to states where
these can function again as originally intended, taking into consideration how each individual object will be restored and the sequence in which the objects will be restored. Indirect costs are those related to loss of connectivity and a deficient level of service, however that is defined for a given network (e.g., temporal prolongation of travel for road networks). Since the spatial and temporal correlation between events are considered, risk can be said to be spatially and temporally distributed (i.e., risk estimates vary in space within a defined area of study, and in time with a defined period of analysis).
Furthermore, network managers may be often interested in investigating the effect of hazard loads on their critical objects. Therefore, managers need to be given the option to select hazard events based on the periodicity of the manifested site-specific hazard loads (i.e., managers may not be as concerned with selecting a hazard event based on the return period of the preceding source event). To meet this need, the presented risk equation (3) can be modified by first selecting the hazard events according to the return periods related to the site-specific loads (i.e., the probability of occurrence of a scenario $\mathbb{P}[E^{\mathrm{src}}|E^{\mathrm{haz}}]$ is determined
by source event $E^{\mathrm{src}}$ leading to an intended hazard event $E^{\mathrm{haz}}$ as defined by Hackl et al. (2017)).

### 4.8.2  Direct costs

Only restoration costs were considered as direct costs. For each section $n$ in a damage state $ds$, a restoration intervention was assigned. Associated with each intervention were (i) the capacity losses due to the execution of the intervention, (ii) the length of time required to execute the intervention $\tau_n \geq 0$, and (iii) the cost of the intervention $c_n \geq 0$. This cost was composed of a
fixed part $c_n^{\mathrm{fix}}$ (e.g., site setup) and a variable part $c_n^{\mathrm{var}}$ (e.g., mu/m$^2$ of pavement, mu/m$^3$ of concrete, where mu stands for



monetary units). Based on the derived fragility functions and the observed time-dependent intensity measures $\Xi_t$, the expected restoration costs $\langle c \rangle_{n,t}$ and durations $\langle \tau \rangle_{n,t}$ for each section were calculated.

$$\langle c \rangle_{n,t} = \mathbb{E}\left(c_n | \Xi_t\right) = \sum_{ds} \mathbb{E}\left(c_n^{\text{fix}} + c_n^{\text{var}} | ds\right) \mathbb{P}[ds|\Xi_t] \tag{4}$$

$$\langle \tau \rangle_{n,t} = \mathbb{E}\left(\tau_n | \Xi_t\right) = \sum_{ds} \mathbb{E}\left(\tau_n | ds\right) \mathbb{P}[ds|\Xi_t] \tag{5}$$

5     The overall expected direct costs $c^{\text{dc}}$ was the sum of the expected direct costs for each intervention executed. It was assumed that intervention costs are not affected by the selected restoration program.

$$c^{\text{dc}} = \sum_n \max_t(\langle c \rangle_{n,t}) \tag{6}$$

    Cost estimates were based on Staubli and Hirt (2005) and from a survey conducted by D'Ayala and Gehl (2015). For each object type and damage state, a restoration strategy was derived, and for each strategy, cost[4] and duration values were 10  approximated (Table 7).

| Event | State | Duration | Fixed costs | Variable costs |
|---|---|---|---|---|
| | | [hour/pier] | [mu] | [mu/pier] |
| bridge local scour | 1 | 60 | 16,000 | 24,000 |
| | 2 | 135 | 30,000 | 40,000 |
| | 3 | 240 | 48,000 | 64,000 |
| | | [hour/m$^2$] | [mu] | [mu/m$^2$] |
| pavement mudblocking | 1 | 0.005 | 3,500 | 16.50 |
| | 2 | 0.009 | 9,600 | 165.00 |
| | 3 | 0.015 | 14,400 | 325.00 |
| | | [hour/m$^2$] | [mu] | [mu/m$^2$] |
| pavement inundation | 1 | 0.005 | 3,500 | 16.50 |
| | 2 | 0.009 | 9,600 | 165.00 |
| | 3 | 0.015 | 14,400 | 325.00 |

**Table 7.** Restoration costs and durations.

---

[4]Costs taken from the literature are adjusted to 2017 price levels. To avoid overinterpreting the specific values that were in the example, monetary units are used instead of real currency.



### 4.8.3 Indirect costs

The indirect costs were comprised of costs for the temporal prolongation of travel and costs due to a loss of connectivity. The overall indirect costs $c^{\text{ic}}$ were measured as the difference between indirect costs at time $t$ and the indirect costs at time 0 when the network was fully functional.

$$c^{\text{ic}} = \sum_t \left[ \sum_{e \in \mathcal{P}^1_{od,t}} C^{f,\text{pt}}(x_{e,t}) + C^{f,\text{lc}}(\mathcal{P}^0_{od,t}) \right] \tag{7}$$

where $C^{f,\text{pt}}$ was a cost function dependent on the edge traffic flow $x_{e,t}$ in time $t$ through edge $e$ that was part of the set of feasible paths $\mathcal{P}^1_{od,t}$ identified in time $t$, and $C^{f,\text{lc}}$ was a cost function dependent on a loss of connectivity, which was determined based on the set of unfeasible paths $\mathcal{P}^0_{od,t}$ identified in time $t$.

*Temporal prolongation of travel*

The cost function attributed to traffic flow included sub-functions to estimate the costs related to travel time $C^{f,\text{tt}}$ and vehicle operation $C^{f,\text{vo}}$.

$$C^{f,\text{pt}}(x_{e,t}) = C^{f,\text{tt}}(x_{e,t}) + C^{f,\text{vo}}(x_{e,t}) \tag{8}$$

*Travel time costs* were estimated based on the increased amount of time people spent traveling, which was linked directly to the flow on an edge.

$$C^{f,\text{tt}}(x_{e,t}) = (t^T_{e,t} \cdot x_{e,t} - t^T_{e,0} \cdot x_{e,0}) \cdot \xi \tag{9}$$

where $t^T_{e,t}$ was the travel time on edge $e$ at time $t$ in hours and $\xi$ was the value of travel time. Based on the work of the Swiss Association of Road and Transport Experts (VSS, 2009a), $\xi$ was assumed to be $23.02\,\text{mu/hour}$ per vehicle.

*Vehicle operation costs* were incurred as a result of fuel consumption and vehicle maintenance.

$$C^{f,\text{vo}}(x_{e,t}) = (x_{e,t} - x_{e,0}) \cdot l_e \cdot (\zeta \cdot F + \rho) \tag{10}$$

where $l_e$ was the length of edge $e$, $\zeta$ was the mean fuel price ($1.88\,\text{mu/liter}$), $F$ was the mean fuel consumption ($6.7\,\text{liter}$ per $100\,\text{km}$ per vehicle), and $\rho$ was the operating cost without fuel ($14.39\,\text{mu}/(100 \cdot \text{veh} - \text{km})$) (VSS, 2009b).

*Loss of connectivity*

The costs due to a loss of connectivity were estimated based on the unsatisfied demand per time $t$ and the resulting costs due to a loss caused of the missed trips.

$$C^{f,\text{lc}}(\mathcal{P}^0_{od,t}) = \sum_{od} \sum_{P \in \mathcal{P}^0_{od,t}} f^f_{od}(P) \cdot \epsilon \tag{11}$$





where $f_{od}^f$ was a function used to estimate the demand on any given path for a specific origin-destination $od$, and $\epsilon$ was the monetary loss due to missed trips (i.e., cost of lost labor productivity per hour), which was assumed to be $83.27\,\mathrm{mu}$ for every time step of simulation during the hazards event period. The missed trip cost during the restoration period was assumed to be $666.16\,\mathrm{mu}$ for every simulation time step.

### 4.8.4 Uncertainty

The simulation engine was used to estimate the probable consequences of 1,200 events of different return periods (i.e., 2, 5, 10, 25, 50, 100, 250, 500, 1,000, 2,500, 5,000, 10,000). Therefore, the main source of uncertainty was expected to be related to the source and hazard events. Further considered uncertainty sources were those associated with the triggering of mudflow events, and the damage and functional loss estimations. Other sources of uncertainty were excluded in this study, but could be easily incorporated if required by further analyses.

Such a set of simulations enabled the generation of risk curves for different percentiles of consequences (i.e., the full distribution function of the costs in the area under consideration). By selecting a risk curve of a specific percentile, it was possible to estimate the corresponding annualized risk.

### 4.9 Results

Figure 4 shows how the rainfall (source event), flood and mudflows (hazard events), damages (infrastructure object events), reduction in speed limits and road capacity (network events) and traffic flow (societal event) change during the occurrence period of the hazard events. All the changes correspond to a single event of a 500-year return period. For this specific chain of events, the hazard events extend over a period of 18 hours, after that, the inspection-restoration period starts.

In this scenario, the precipitation period starts at hour 1 and ends at hour 10. The precipitation moves from the northwestern part of the study area to southeastern part, the maximum precipitation can be observed around hours 6 and 7. At hour 6, parts of the motorway to Fürstenau are flooded, which lead to a detour of the traffic through the village of Bonaduz. At hour 8, the motorway is flooded between Tamins and Domat/Ems, which interrupts the west-east connection in the valley. Traffic has to detour through Chur and drive south.

In addition to the evolution of the flood event over time, the triggering of different mudflow events can be observed. For example, in hour 10, a landslide next to the village of Felsberg is triggered, causing the blockage of two minor roads. Due to the lag runoff, the maximum extent of the flood event is reached at hour 11. Most of the flood damage is caused in the western part of Chur, while in the northern part, only the motorway and several minor roads are flooded.

While Figure 4 shows the spatio-temporal progression of one specific scenario, Figure 5 illustrates the aggregated simulation results for 100 scenarios with 500-year return periods. On the top left, the 25, 50 and 75-percentile precipitation fields are shown, with darker areas indicating more intense rainfall. The hazard events of interest are also presented, specifically the 5, 50 and 95-percentile of possible inundation depths as well as the location of possible mudflows color-coded according to their probability of occurrence. The expected discharge along the river is also illustrated in the graph. It can be observed that the



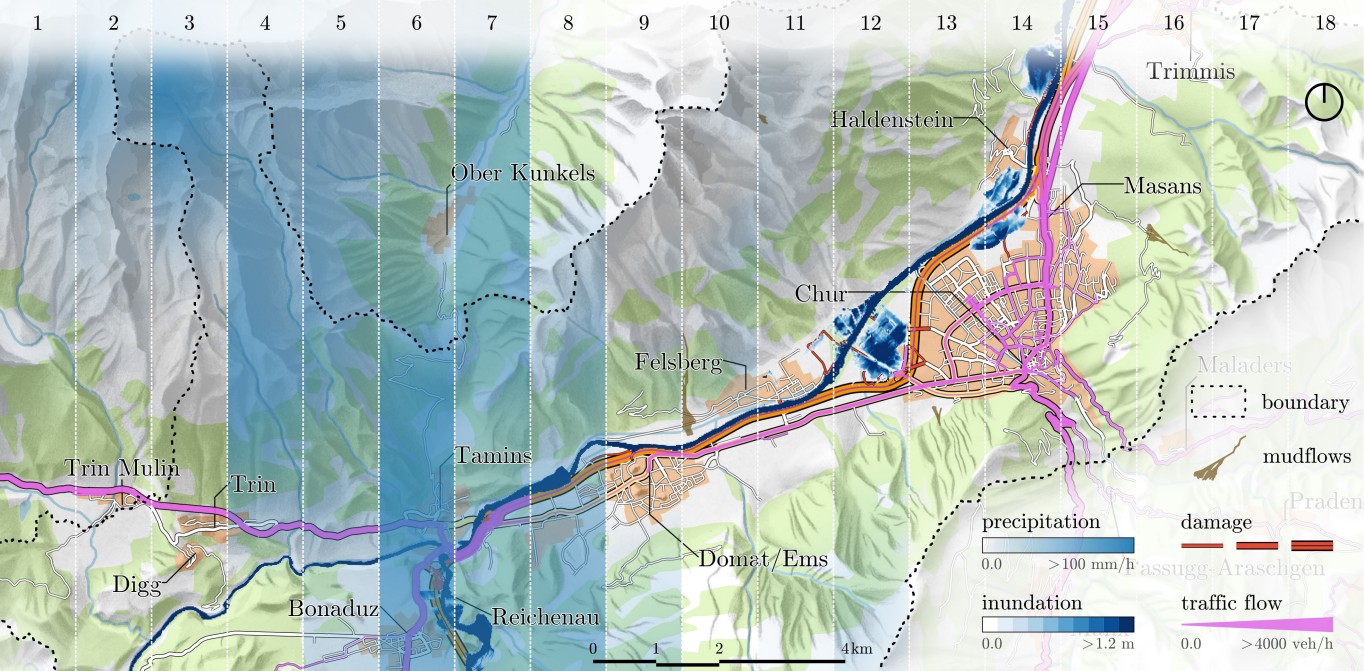

**Figure 4.** Spatio-temporal system result.

5, 50 and 95-percentile values are approximately $1{,}690\,\mathrm{m^3/s}$ at section 30. This occurs because this value corresponds to the targeted discharge value for a 500-year flood event at the predefined gauging station located in that section.

For the 100 scenarios with 500-year return periods, the cumulative direct (restoration) and indirect (traffic) costs during the hazards event period (i.e., as damages occur), with the indirect costs reflecting the total cost of additional travel time and missed
trips, are shown in Figure 6. These costs are represented by the median costs, the 50% and the 90% confidence interval. In both cases, uncertainty is observed to increase as the hazard events periods continues. While the uncertainty of the indirect cost is symmetrical, the uncertainty of the direct cost is heavy-tailed, indicating that for a few scenarios, the costs can be multiple times the expected costs.

Performing the analysis for multiple return periods, as indicated in Section 4.8.4, allows to plot risk curves (i.e., cost versus
return period). These are presented in Figure 7. It is also possible to plot similar curves, such as restoration time versus return period period, as shown in Figure 8. The expected restoration time varies between 0 and 170 days, showing an increasing trend with the severity of the hazard events.

## 5   Discussion

Since the main goal of the presented example was to illustrate the application of the methodology, and therefore, to give the
reader insights into all stages of the risk assessment, models were selected based on the data available and a desire to keep





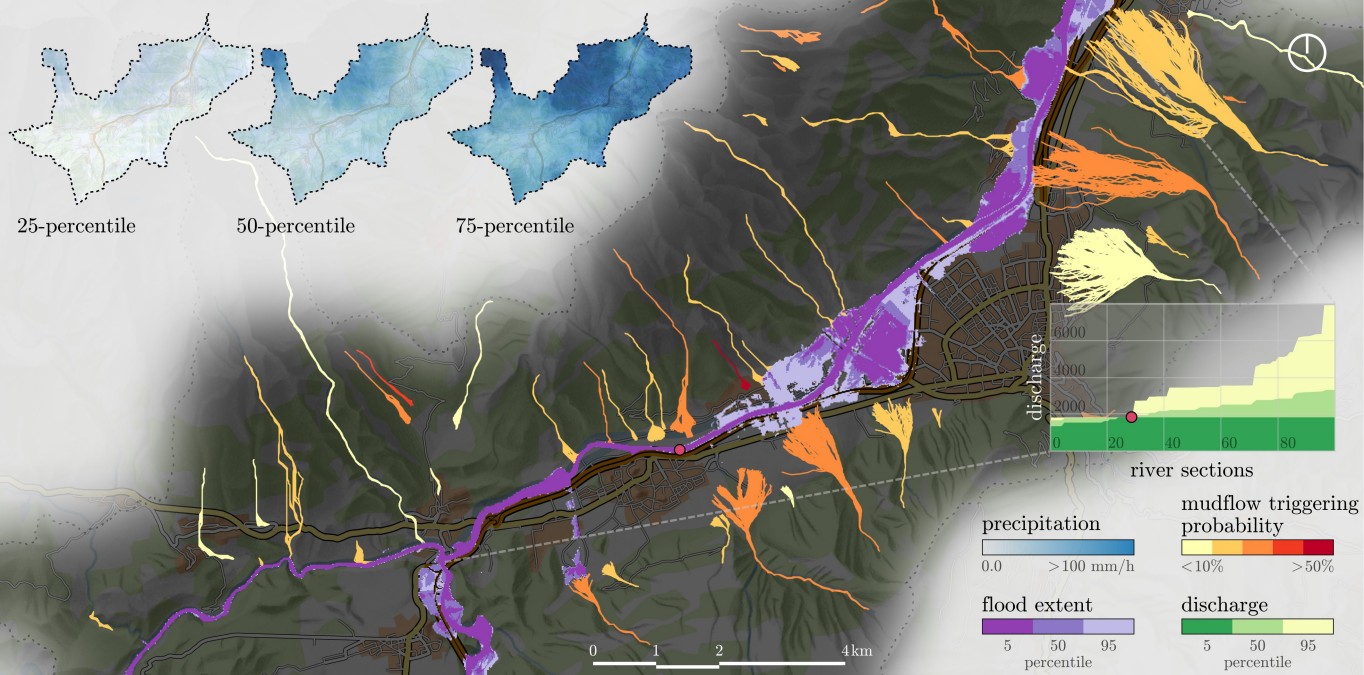

**Figure 5.** Aggregated simulation.

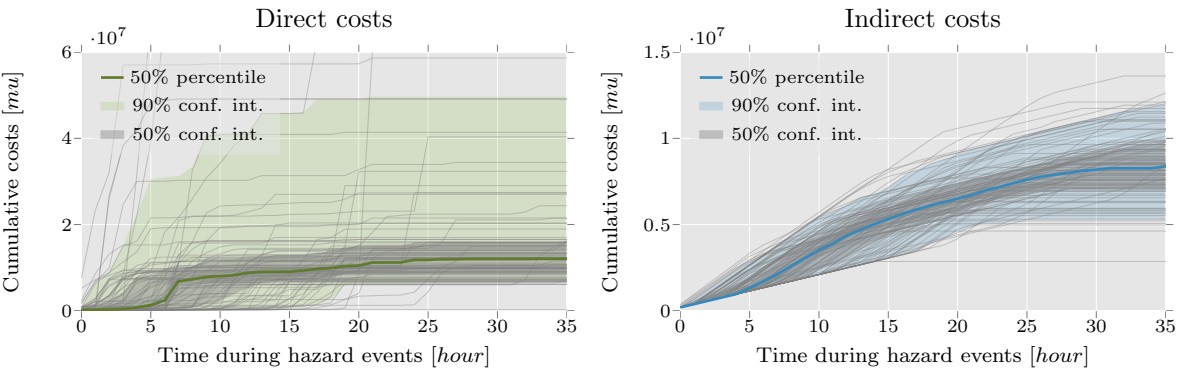

**Figure 6.** Cumulative costs during the hazard events.

computational time low to support the exploration of a large set of scenarios. Multiple hazards of various return periods can be considered along with changes in the traffic flow and restoration activities, leading to a more encompassing way to estimate risk.

Due to the modular approach used, if desired, more sophisticated models may be integrated in the future. For example, the traffic model used here provided a low-complexity representation of driver behavior (e.g., travelers had full knowledge of the traffic conditions). Moreover, the traffic model did not account for dynamic phenomena like queues, spillbacks, wave propa-




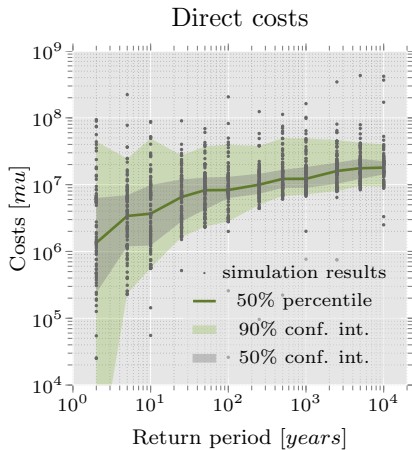
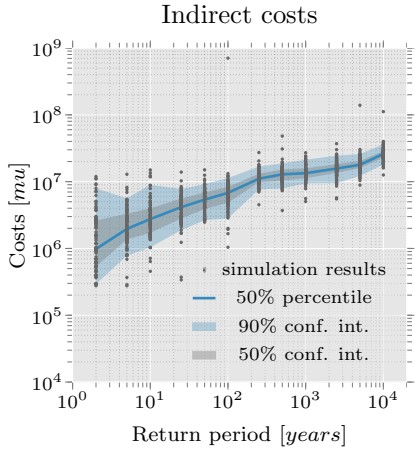
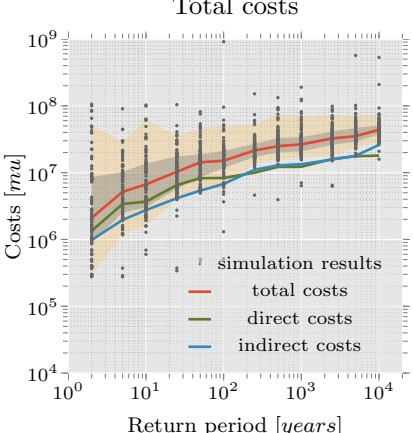

**Figure 7.** Cost vs return period.

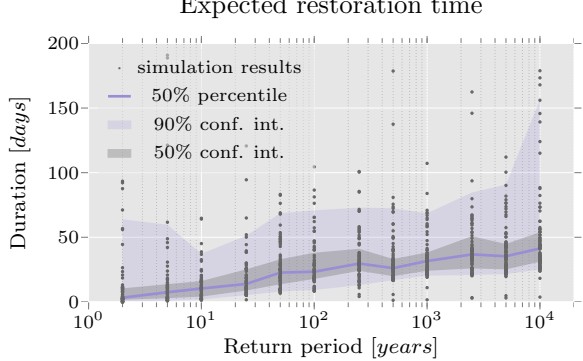

**Figure 8.** Expected restoration time.

gation, or changes in travel patterns after a disruptive event, although studies show these can be considerably different after a disruptive event (e.g., Chang and Nojima (2001); Kontou et al. (2017)). In general, the integration of more sophisticated models can potentially lead to improved risk estimates. A prerequisite for this integration is conducting uncertainty and sensitivity analyses to prioritize the parts of the system to be analyzed in more detail.

5 Because of the modular structure and the universal nature of the models used here, however, the implemented simulation engine is transferable to other river systems and road networks, provided the required data sets are available. The quantity and quality of data needed for the simulation engine are at a reasonable level. In fact, data used in this study are largely available for many locations around the world, enabling the use of the engine in other contexts.

Alternatively, single parts of the simulation engine may be applied independently (e.g., to investigate the probability of
10 bridge failure due to local scour at a given location). Hence, the system may be profitably used for a number of additional





purposes (e.g., as a tool for cost-benefit analysis of flood and mudflow protection measures, as a decision support system for operational flood and mudflow control).

As implemented, the risk assessment can provide a mechanism for a region-wide screening of priority locations for risk reduction based on the analysis of the road network and traffic properties (e.g., please see Heitzler et al. (2017)). This function

can be enhanced with the use of visualization tools, enabling network managers to dynamically see how different environmental systems (i.e., rainfall, flood, mudflow) influence each other, and how their impacts on networks can affect society (Heitzler et al., 2016).

Finally, combining several models results in a significant degree of uncertainty. Whenever possible, the results should be compared with and calibrated against empirical data, when available. For example, the probability of physical damages ob-

tained through simulations could be calibrated against collected data from field structural surveys when such data exist. For societal effects, however, calibration is rather difficult because such data are hard to measure and monitor. Nonetheless, in some cases, basic data are available, including the duration of a network's loss functionality and the estimated number of network users affected.

## 6   Conclusions

The purpose of estimating the risk related to networks is, among others, to provide an overview of probable adverse events that may negatively affect the network, assess societal effects (e.g., costs), and provide a basis for planning risk reducing interventions. Assessing the risk related to networks exposed to multiple types of hazards is not simple, however. This is particularly evident when dealing with complex system representations, where the costs of indirect consequences can be multiple times higher than the costs of direct consequences, with no linear relationship between these types of cost.

This paper describes an innovative risk assessment methodology for networks when there is a need to represent the system containing these networks using various degrees of complexity. The proposed methodology is designed to be a supplement to already existing risk management processes that also support the assessment of the network related risks.

Following this methodology, a state of the art example was conducted, which aimed at estimating the spatio-temporal risk of a road network in Switzerland due to the occurrence of a time-varying rainfall event that caused a flood event and mudflow

events. To achieve this objective, a modular approach was used to couple rainfall, runoff, flood, mudflow, physical damages, functional loss, traffic, and restoration modeling. Consequences were monetized into direct and indirect costs, considering restoration interventions, prolongation of travel, and lost trips.

The use of the methodology is not limited to hydrometeorological events or even road networks. For example, the methodology can be applied to other natural hazards (e.g., earthquakes (Lam and Adey, 2016), coastal floods, rockfalls) or other

networks (e.g., railway networks (Papathanasiou et al., 2016), waterways, and even inter-modal studies). Such system-wide analyses would require appropriate hazard models and descriptors of the relationships between the hazard events, physical damages and functional disruptions (e.g., no rail traffic due to settlement of the rail tracks conditional on the earthquake event).





The application of the risk assessment methodology can result in a data intensive risk assessment design, depending on the approach selected and the complexity of the system representation. This may increase the time required to compute risk estimates, particularly if the area of study does not have the needed spatial data sets for the models selected.

# 7  Acknowledgements

5    The work presented here has received funding from the European Union's Seventh Programme for Research, Technological Development and Demonstration under grant agreement No 603960, and from Horizon 2020, the European Union's Framework Programme for Research and Innovation, under grant agreement No 636285.

# Appendix A:  Notation

*General*

$t$ ........ time step

$g$ ....... gravitational acceleration

$\mathbb{E}$ ....... expected value

$\mathbb{P}$ ....... probability

$\mathcal{R}$ ....... risk

$E^{\mathrm{haz}}$ .... hazard event

$E^{\mathrm{src}}$ .... source event

$E^{\mathrm{inf}}$ .... infrastructure object event

$E^{\mathrm{net}}$ .... network event

$E^{\mathrm{soc}}$ .... societal event

*Geodata*

$W$ ...... study area

$\boldsymbol{x}$ ....... point in $W$

$r$ ........ river station

$n$ ....... a section

$e$ ........ an infrastructure object (i.e., network edge)

$\mathcal{E}$ ....... set of all infrastructure object

$\boldsymbol{c}$ ....... a raster cell

$\boldsymbol{c}_r$ ...... raster cell located at the watershed outlet corresponding to a river station $r$

$\boldsymbol{c}_n$ ...... raster cell where section $n$ is located

$\boldsymbol{c}_\ell$ ....... raster cell where potential mudflow location $\ell$ may be

*Rainfall model*

$\tau^{\mathrm{src}}$ ..... duration of rainfall event

$\boldsymbol{P}_{\tau^{\mathrm{src}}}$ .... time series of precipitation fields of duration $\tau^{src}$

$\boldsymbol{P}_t$ ...... precipitation field for time $t$

$p_{\boldsymbol{c},t}$ ..... precipitation value at cell $\boldsymbol{c}$ for time $t$

*Runoff model*

$p_{\boldsymbol{c},t}^e$ ..... accumulated precipitation excess at cell $\boldsymbol{c}$ for time $t$

$\mathrm{CN}_{\boldsymbol{c}}$ .... curve number for cell $\boldsymbol{c}$

$t^{\mathrm{wc}}$ ...... time of concentration for the watershed

$t_{\boldsymbol{c}}^{\mathrm{wc}}$ ...... lag time of travel for cell $\boldsymbol{c}$

$d^{\mathrm{wc}}$ ..... travel distance from the cell that is most distant from the watershed outlet to the watershed outlet

$d_{\boldsymbol{c}}^{\mathrm{wc}}$ ..... travel distance from cell $\boldsymbol{c}$ to the watershed outlet

$f_{\boldsymbol{c},t}^{\mathrm{out}}$ ..... outflow of cell $\boldsymbol{c}$ for time $t$

$f_{\boldsymbol{c},t}^{\mathrm{in}}$ ..... average inflow for cell $\boldsymbol{c}$ for time $t$

$\varrho$ ....... storage coefficient for linear reservoirs

$\Delta t^Q$ .... time interval of a hydrograph $Q$

$Q_{r,t}$ ..... flow for a river station $r$ at time $t$

$Q_{r,0}$ .... base flow for a river station $r$

*Flood model*





$i$ ........ index for cross section

$L_{i,i+1}$ ... channel reach length between cross-section $i$ and cross-section $i+1$

$z_i$ ....... bed elevation with regard to the datum at cross-section $i$

$h_{i,t}$ ..... water depth at cross-section $i$ at time $t$

$\gamma_i$ ....... energy correction factor at cross-section $i$

$v_{i,t}$ ...... average flow velocity at cross-section $i$ at time $t$

$\bar{S}_{i,i+1}$ ... average friction slope between cross-section $i$ and cross-section $i+1$

$A_{i,t}$ ..... wetted cross-sectional area at cross-section $i$ at time $t$

$b_i$ ....... channel width at cross-section $i$

$\boldsymbol{I}_t$ ....... inundation field at time $t$

*Mudflow model*

$p^{\tau}_{\boldsymbol{c}_\ell,t}$ .... precipitation threshold for intensity-duration function at cell $\boldsymbol{c}_\ell$ and time $t$

$\tau_{\boldsymbol{c}_\ell,t}$ ..... duration of the rainfall event up until time $t$ at cell $\boldsymbol{c}_\ell$

$FS_{\ell,t}$ ... factor of safety for mudflow $\ell$ at time $t$

$c^{s+r}$ .... cohesion of soil and roots

$c^s$ ....... cohesion of soil

$c^r$ ....... cohesion of roots

$\gamma^s$ ...... specific weight of soil

$\gamma^w$ ...... specific weight of water

$m_t$ ...... fraction between water table depth and the soil depth at time $t$

$z^s$ ....... soil depth

$z^w_t$ ...... water table depth at time $t$

$S$ ....... slope angle

$\phi$ ....... angle of internal friction

$\mathbb{P}[\ell|t]$ .... probability of mudflow $\ell$ triggered at time $t$

$V_\ell$ ...... volume of mudflow $\ell$

$R_\ell$ ...... runout length of mudflow $\ell$

$\boldsymbol{L}_t$ ...... mudflow elevation field at time $t$

*Bridge local scour model*

$h^{\text{sc}}_{e,r,t}$ .... scour depth at time $t$ for object $e$ located near river station $r$

$\kappa$ ....... corrective coefficients

$a_e$ ...... pier width for object $e$

$DS$ ..... realization of a damage state

$ds$ ...... damage (limit) state

$\Xi_t$ ...... hazard intensity at time $t$

*Functional loss model for bridge local scour and pavement*
*Mudblocking*

$\Xi_t$ ...... hazard intensity at time $t$

$ds$ ...... damage (limit) state

$\lambda_n$ ...... functional capacity loss for section $n$

$\langle\lambda\rangle_{n,t}$ ... expected functional capacity loss for section $n$ at time $t$

*Functional loss model for pavement inundation*

$v^T_{n,t}$ ..... maximum acceptable velocity for traffic through section $n$ at time $t$

$v^{\max}$ .... maximum allowed speed

$i_{\boldsymbol{c}_n,t}$ .... inundation depth at cell $\boldsymbol{c}_n$ at time $t$

*Functional loss model for the network*

$\lambda_{e,t}$ ..... functional capacity loss for edge $e$ at time $t$

$\langle\lambda\rangle_{e,t}$ ... expected functional capacity loss for edge $e$ at time $t$

$\phi_{e,t}$ ..... relative loss of speed for edge $e$ at time $t$

*Traffic Model*

$x_{e,t}$ ..... traffic flow through edge $e$ at time $t$

$Z^T$ ..... objective function related to traffic

$C^{f,T}$ .... function to estimate travel cost

$P$ ....... a path

$\mathcal{P}^1_{od,t}$ .... set of feasible paths at time $t$

$\mathcal{P}^0_{od,t}$ .... set of unfeasible paths at time $t$

$od$ ...... origin-destination

$f^f_{od}(P)$ .. function to estimate traffic flow on path $P$ that connects origin-destination $od$





$t^T_{e,t}$ ...... travel time through edge $e$ in time $t$

$t^T_{e,0}$ ..... initial free flow travel time

$y_{e,0}$ ..... initial edge capacity

$\alpha_e, \beta_e$ ... calibration parameters for the traffic through edge $e$

*Direct costs*

$c^{\text{dc}}$ ...... direct costs

$C^{f,\text{dc}}$ ... direct cost function

$c_n$ ...... cost of the intervention for section $n$

$c^{\text{fix}}_n$ ...... fixed costs for section $n$

$c^{\text{var}}_n$ ..... variable costs for section $n$

$\langle c \rangle_{n,t}$ ... expected costs of the intervention for section $n$ at time $t$

$\tau_n$ ...... duration of the intervention for section $n$

$\langle \tau \rangle_{n,t}$ ... expected duration of the intervention for section $n$ at time $t$

*Indirect costs*

$c^{\text{ic}}$ ...... indirect costs

$C^{f,\text{ic}}$ .... indirect cost function

$C^{f,\text{pt}}$ .... function for costs of temporal prolongation of travel

$C^{f,\text{lc}}$ .... function for costs due to a loss in connectivity

$C^{f,\text{tt}}$ .... function for costs of travel time

$C^{f,\text{vo}}$ ... function for costs of vehicle operation

$od$ ...... origin-destination

$P$ ....... a path

$\mathcal{P}^1_{od,t}$ .... set of feasible paths at time $t$

$\mathcal{P}^0_{od,t}$ .... set of unfeasible paths at time $t$

$f^f_{od}(P)$ .. function to estimate traffic flow on path $P$ that connects origin-destination $od$

$x_{e,t}$ ..... traffic flow through edge $e$ at time $t$

$t^T_{e,t}$ ...... travel time through edge $e$ in time $t$

$\xi$ ........ value of travel time

$l_e$ ....... length of edge

$F$ ....... mean fuel consumption

$\zeta$ ....... mean fuel price

$\rho$ ....... operating costs without fuel

$\epsilon$ ....... value of a lost trip

## Appendix B: Models used

### B1 Geodata

To represent geospatial elements, including infrastructure objects and natural phenomena, within a computational environment, an entity-based (vector) approach and a continuous field (raster) approach were used. First, the entity-based approach views

5    space as a place to be populated by entities with clearly defined spatial boundaries and associated properties (e.g., an edge within a road network and its type of use). Second, the continuous field approach typically represents natural phenomena as a set of spatially varying values of some attribute, such as precipitation or elevation.

     Various geodata from different sources were necessary to construct and operate the different models implemented. Many of these were extracted from the VECTOR25 dataset (Swisstopo, 2015), the main source for topographic maps of Switzerland

10    with a scale denominator of 25,000. A digital terrain model (DTM) of $16\,\text{m} \times 16\,\text{m}$ resolution based on the DTM Amtliche Vermessung was used to incorporate elevation data in this application. All geodata were given or transformed in the Swiss coordinate reference system CH1903/LV03 (EPSG code:21781).





Three different ways to address the position in space were used: (i) continuous Cartesian coordinates to represent any point $x$ in the study area $W$, (ii) raster coordinates to represent any cell $c \in W$, and (iii) linear coordinates to represent, for example, stations $r \in W$ along the watercourses.

## B2   Rainfall model

The first part of this process was choosing the time series of precipitation fields $\boldsymbol{P}_{\tau^{src}}$ to be used in a given simulation from the precipitation catalog of Wüest et al. (2010). This involved two steps: (i) setting the beginning of the rainfall event from this catalog using a simple random sampling algorithm, and (ii) selecting the duration of the rainfall event $\tau^{src}$.

The latter was accomplished using a simple random sampling algorithm on a scaled Beta probability distribution representing possible duration lengths, ranging from 1 to 72 hours. Each return period of interest had an assigned Beta probability distribution, with larger durations to be observed with higher frequency when modeling events of larger return periods. After these random selections, there were precipitation fields $\boldsymbol{P}_t$, corresponding to each time $t \in \tau^{\mathrm{src}}$.

To further characterize a rainfall event, a second set of actions was needed to relate that event with a given return period. The precipitation values $p_{c,t} \in \boldsymbol{P}_t$ for each raster cell $c$ at time $t$ were iteratively scaled as described in Hackl et al. (2017) until the rainfall event generated a discharge value at a point of interest matching that of the desired return period. The result of this entire process was a time series of scaled precipitation fields $\boldsymbol{P}_{\tau^{\mathrm{src}}}$.

Finally, in order to match the spatial resolution to be used throughout the entire analysis (set at $16\,\mathrm{m} \times 16\,\mathrm{m}$), the resolution of all precipitation fields $\boldsymbol{P}_t$ (originally set at $1\,\mathrm{km} \times 1\,\mathrm{km}$), was adapted using a regridding process.

## B3   Runoff model

The precipitation excess was computed for each cell using the Soil Conservation Service (SCS) Curve Number (CN) model. This model estimates precipitation excess as a function of cumulative precipitation, soil cover, land use, and antecedent moisture general watershed data based on an empirical equation (Feldman, 2000):

$$p_{c,t}^e = \frac{(\mathrm{CN}_c \cdot (p_{c,t} + 50.8) - 5080)^2}{\mathrm{CN}_c \cdot (\mathrm{CN}_c \cdot (p_{c,t} - 203.2) + 20320)} \tag{B1}$$

where $p_{c,t}^e$ is the accumulated precipitation excess for cell $c$ at time $t$, $p_{c,t}$ is the corresponding precipitation value, and $\mathrm{CN}_c$ is the curve number for the cell.

Each cell's excess was then lagged to the basin outlet according to the cell's travel time. This translation time to the outlet was computed through a grid-based travel-time model:

$$t_c^{\mathrm{wc}} = t^{\mathrm{wc}} \cdot \frac{d_c^{\mathrm{wc}}}{d^{\mathrm{wc}}} \tag{B2}$$

where $t_c^{\mathrm{wc}}$ is the lag time of travel for a cell $c$, $t^{\mathrm{wc}}$ is the time of concentration for the watershed, $d_c^{\mathrm{wc}}$ the travel distance from cell $c$ to the watershed outlet, and $d^{\mathrm{wc}}$ the travel distance for the cell that is most distant from the watershed outlet.





The individual cell outflows $f_{c,t}^{\text{out}}$ were routed through a linear reservoir, to account for the effects of watershed storage. The routing was done based on Clark's original methodology:

$$f_{c,t}^{\text{out}} = \frac{2 \cdot \Delta t^Q \cdot (f_{c,t}^{\text{in}} - f_{c,t-1}^{\text{out}})}{2\varrho + \Delta t^Q} + f_{c,t-1}^{\text{out}} \tag{B3}$$

where $f_{c,t}^{\text{in}}$ is the average inflow to the storage of cell $c$ at time $t$ composed of the accumulated precipitation excess $p_{c,t}^e$ and the outflows of the neighbor cells at $t-1$, $\varrho$ is a storage coefficient for linear reservoirs (defined in time units), and $\Delta t^Q$ is the time interval of a hydrograph $Q$ (here set to $1$ hour).

The results from each cell were combined to produce the final hydrographs for each river station $r$ using the corresponding estimated flows $Q_{r,t}$ for all time steps $t$. These flows were estimated by adding the outflow values $f_{c_r,t}^{\text{out}}$ of the cells $c_r$ located at the watershed outlet that corresponds to the river station of interest $r$, and the base flow $Q_{r,0}$ of that station:

$$Q_{r,t} = Q_{r,0} + \sum_{c_r} f_{c_r,t}^{\text{out}} \tag{B4}$$

**B4   Flood model**

The governing equation describing the flow problem of the one-dimensional hydraulic model was derived by the energy equation for two neighboring cross-sections, enclosing a channel reach of length $L_{i,i+1}$:

$$z_i + h_{i,t} + \frac{\gamma_i \cdot v_{i,t}^2}{2g} = z_{i+1} + h_{i+1,t} + \frac{\gamma_{i+1} \cdot v_{i+1,t}^2}{2 \cdot g} + \bar{S}_{i,i+1} \cdot L_{i,i+1} \tag{B5}$$

where $z_i$ is the bed elevation with regard to the datum, $h_{i,t}$ is the water depth at time $t$, $\gamma_i$ is the energy correction factor, and $v_{i,t}$ is the average flow velocity at time $t$, with all of these variables for a given cross-section $i$. Moreover, $g$ is the gravitational acceleration, $\bar{S}_{i,i+1}$ is the average friction slope between both cross-sections, index $i$ denotes an upstream cross-section, and index $i+1$ denotes a downstream cross-section. The friction slope can be calculated based on different empirical laws (e.g., the Manning formula). The average flow velocity $v_{i,t} = Q_{i,t}/A_{i,t}$ can be expressed as a function of the discharge $Q_{i,t} = Q_{r,t}$ and the wetted cross-sectional area $A_{i,t}$. At the same time, for a given cross-section $i$, this area $A_{i,t} = h_{i,t} \cdot b_i$ can be expressed as a function of the water depth $h_{i,t}$ at time $t$ and the width of the channel $b_i$. Equation (B5) allows to compute the water surface profiles from one cross-section to the next. For most cases, this has to be done numerically. Finally, the water depth $h$ values at each river cross-section were interpolated to obtain an inundation field $I_t$, representing a raster file for time $t$.





## B5 Mudflow model

Potential mudflow locations $\boldsymbol{c}_\ell \in W$ were obtained from Losey and Wehrli (2013). The probability that a mudflow could occur was estimated based on precipitation thresholds obtained by using the empirical intensity-duration function for sub-alpine regions of Zimmermann et al. (1997):

$$p^\tau_{\boldsymbol{c}_\ell,t} = 32 \cdot \tau^{-0.72}_{\boldsymbol{c}_\ell,t} \tag{B6}$$

where $p^\tau_{\boldsymbol{c}_\ell,t}$ is the precipitation threshold in $\mathrm{mm/hour}$ and $\tau_{\boldsymbol{c}_\ell,t}$ is the duration of the rainfall event up until time $t$ at the potential mudflow location $\boldsymbol{c}_\ell$. For each potential mudflow location, the respective precipitation values $p_{\boldsymbol{c}_\ell,t}$ were extracted and used as points of comparisons. If the threshold was exceeded ($\sum_{t \in \tau_{\boldsymbol{c}_\ell,t}} p_{\boldsymbol{c}_\ell,t} > p^\tau_{\boldsymbol{c}_\ell,t}$) at a given time step, a probability of being triggered was assigned to the event, based on the slope factor of safety (FS) (Skempton and Delory, 1952):

$$FS_{\ell,t} = \frac{(c^s + c^r) + (\gamma^s - m^t \cdot \gamma^w) \cdot z^s \cdot \cos^2 S \cdot \tan \phi}{\gamma^s \cdot z^s \cdot \sin S \cdot \cos S} \tag{B7}$$

where $c^s$ and $c^r$ are the cohesion of soil and roots respectively, $\gamma^s$ is the specific weight of soil, $m^t = z^w_t/z^s$ is the fraction between water table depth $z^w_t$ at time $t$ and the soil depth $z^s$, $\gamma^w$ is the specific weight of water, $S$ is the slope angle, and $\phi$ is the angle of internal friction. The water table depth $z^w_t$ is composed of the initial water table depth $z^w_0$ and the additional depth $\sum_{t \in \tau_{\boldsymbol{c}_\ell,t}} p_{\boldsymbol{c}_\ell,t}$. All values can be assumed to correspond to the potential mudflow location $\boldsymbol{c}_\ell$. Based on probabilistic
input parameters (please see Table B1), a Monte Carlo scheme was used to generate $j = 100{,}000$ FS values. This data set was then used to derive the triggering probability ($\mathbb{P}[\ell|t] = \frac{1}{j}\sum_j \mathbb{1}_{FS_{\ell,t}<1}$).

| Sym. | Description | Distr. | Values | Unit |
|---|---|---|---|---|
| $c^s$ | cohesion of soil | Norm | 5.04, 2.18 | kPa |
| $c^r$ | cohesion of roots | Norm | 3.41, 2.36 | kPa |
| $\gamma^s$ | specific weight of soil | Unif | 18, 33 | kN/m³ |
| $\gamma^w$ | specific weight of water | Det | 9.81 | kN/m³ |
| $z^s$ | soil depth | Unif | 0.1, 1.5 | m |
| $S$ | slope angle | Unif | 35, 65 | Deg |
| $\phi$ | angle of internal friction | Norm | 30, 5 | Deg |

**Table B1.** Probabilistic inputs for mudflow triggering.

The volume $V_\ell$ of each mudflow was estimated by taking into account the runout length $R_\ell$ of the fan using an empirical relation (Rickenmann, 1999):

$$V_\ell = \left(\frac{R_\ell}{15}\right)^2 \tag{B8}$$



The increase in elevation per cell was calculated by dividing the volume by the area of the fan. The output of the model was a time series of raster files $L_t$, whose cell values corresponded to the additional elevation caused by the mudflows.

## B6 Bridge local scour model

Empirical relationships from Arneson et al. (2012) were used to quantify the excavated depth $h^{\text{sc}}_{e,r,t}$ of an object $e$ located near
river station $r$ due to local scour at time $t$:

$$h^{\text{sc}}_{e,r,t} = 2.0 \cdot \kappa_1 \cdot \kappa_2 \cdot \kappa_3 \cdot h_r \cdot \left( \frac{a_e}{h_{r,t}} \right)^{0.65} \left( \frac{v_{r,t}}{\sqrt{g \cdot h_{r,t}}} \right)^{0.43} \tag{B9}$$

where the $\kappa$ parameters are corrective coefficients and $a_e$ represents the pier width. The relationship between the water depth $h_{r,t}$, flow velocity $v_{r,t}$ and discharge $Q_{r,t}$ is given in Section B4. Based on probabilistic input parameters, a Monte Carlo scheme was implemented to generate 100,000 scour depths $h^{\text{sc}}_{e,r,t}$. This dataset was then entered into a maximum likelihood
estimation function to generate fragility functions for the four damage states given in Table 2 with respect to flow discharge $Q_{r,t}$. The functions followed log-normal relationships:

$$\mathbb{P}[DS \geq ds | \Xi_t] = \Phi \left( \frac{\ln \Xi_t - \mu}{\sigma} \right) \tag{B10}$$

where $DS$ represents the realization of the damage state to be compared against a threshold damage state $ds$, and $\Xi_t$ represents the hazard intensity measure at time $t$. The derived fragility parameters for the scour damage states are given in Table B2.

| State | Label | 1-pier bridge | | 2-pier bridge | |
|:---:|:---|:---:|:---:|:---:|:---:|
| | | $\mu$ | $\sigma$ | $\mu$ | $\sigma$ |
| 0 | *operational* | - | - | - | - |
| 1 | *monitored* | 5.52 | 0.66 | 4.67 | 0.64 |
| 2 | *capacity-reduced* | 8.34 | 0.58 | 7.44 | 0.54 |
| 3 | *closed* | 10.1 | 0.61 | 9.10 | 0.54 |

**Table B2.** Fragility function parameters for bridge local scour.

## 15   B7   Pavement inundation model

The fragility functions were constructed assuming that (i) the general width of high-speed (major) roads was $12\,\text{m}$ and that of local (minor) roads was $6\,\text{m}$, (ii) all pavements had a sub-base, with major roads having a sub-base twice as thick as that of minor roads, (iii) major road layers were considered to always be thicker than local road layers, (iv) one day of inundation could compromise the performance of the subgrade layer (Roslan et al., 2015), and (v) any amount of traffic on a road section
with a compromised subgrade layer would result in reconstruction. Log-normal fragility functions were fitted based on three additional assumptions:



- the sub-base of a linear meter of major road section can store $0.35\,\mathrm{m}^3$ of water (Walsh, 2011), leading to assume that inundation depths below $2.92\,\mathrm{cm}$ caused problems to major road sections with 5 percent probability (the same threshold for minor road sections was set to $1.46\,\mathrm{cm}$),

- an inundation depth of $30\,\mathrm{cm}$ is the average depth at which passenger cars start to float, which implies that objects as heavy as passenger cars can be transported throughout the road network, leading to assume the collapse of the drainage function and significant damages to various road elements in addition to making the subgrade vulnerable with 95 percent probability,

- the median inundation depth values of the fragility functions arbitrarily increase by $5\,\mathrm{cm}$ as the damage states increase, with median values of major roads higher by $10\,\mathrm{cm}$ than those of local roads to illustrate that pavement thickness is

a vulnerability factor as indicated by Zhang et al. (2008) and acknowledge that major roads undergo a more rigorous design process than local roads.

Table B3 shows the parameters of the fragility function when inundation depth was used as an intensity measure. Such depth was associated with the need to clean up a given road section, damages to selected elements, and the eventual loss of the subgrade.

| State | Label | major road | | minor road | |
|---|---|---|---|---|---|
| | | $\mu$ | $\sigma$ | $\mu$ | $\sigma$ |
| 0 | *operational* | - | - | - | - |
| 1 | *monitored* | 2.71 | 0.74 | 1.61 | 0.92 |
| 2 | *capacity-reduced* | 3.00 | 0.87 | 2.30 | 0.93 |
| 3 | *closed* | 3.22 | 1.00 | 2.71 | 1.00 |

**Table B3.** Fragility function parameters for pavement inundation (when using inundation depth as intensity measure).

Other modes of failure, in particular the blockage of drainage, delamination, erosion and washed out elements were associated with runoff flow. Although important to model, these phenomena were not included in the model, but should certainly be considered in the future.

## B8 Pavement mudblocking model

As part of a survey conducted by Winter et al. (2013), experts assigned damage state exceedance probabilities to debris flow

volumes for specific damage states and road categories (i.e., major roads and minor roads). Volumes were understood to intersect a road section of $500\,\mathrm{m}$. Experts also provided a score representing their level of expertise.

This dataset was used to derive fragility functions for pavement mudblocking. For every combination of damage state and road category, four expert responses were randomly sampled from the survey dataset. This process resulted in different scenarios of relationships between debris flow volumes and damage state exceedance probabilities. These sampled relationships,

along with the recorded expertise level scores, were entered into a maximum likelihood estimation function to generate the fragility functions given in Table B4.



| State | Label | major road | | minor road | |
|---|---|---|---|---|---|
| | | $\mu$ | $\sigma$ | $\mu$ | $\sigma$ |
| 0 | *no damage* | - | - | - | - |
| 1 | *limited damage* | 9.07 | 3.45 | 7.70 | 3.70 |
| 2 | *serious damage* | 10.6 | 2.92 | 9.28 | 3.31 |
| 3 | *destroyed* | 11.5 | 2.28 | 9.79 | 2.51 |

**Table B4.** Fragility function parameters for pavement mudblocking.

## B9 Functional loss model for bridge local scour and pavement mudblocking

Please see Section 4.7.8

## B10 Functional loss model for pavement inundation

Please see Section 4.7.9

## B11 Functional loss model for the network

An aggregation routine of sections' functional losses was implemented, which computed the expected functional loss of an edge level by identifying the maximum expected functional loss of the sections that are part of the edge. The functional loss related to road capacity for an edge $e$ at time $t$, was determined by:

$$\langle \lambda \rangle_{e,t} = \max_{n \in e} (\langle \lambda \rangle_{n,t}) \tag{B11}$$

At the same time, the functional loss due to speed reduction for an edge $e$ at time $t$, was determined by:

$$\phi_{e,t} = \max(0, v_{e,t} - \max_{n \in e}(v_{n,t}^T))/v_{e,t} \qquad \phi_{e,t} \in [0,1] \tag{B12}$$

where $\phi_{e,t}$ is the relative loss of speed, and $v_{e,t}$ is the maximum allowed speed, both for object $e$ at time $t$.

## B12 Traffic model

The traffic flow $x_{e,t}$ for edge $e$ at time $t$ was estimated by solving the user equilibrium assignment, Eq. (B13a) subjected to Eqs. (B13b) and (B13c).

$$x_{e,t} \in \min Z^T = \sum_e \int_0^{x_{e,t}} C^{f,T}(\omega)\,d\omega \tag{B13a}$$





subject to

$$\sum_{P \in \mathcal{P}^1_{od,t}} f^f_{od}(P) = d_{od} \qquad\qquad \forall od \quad \text{(B13b)}$$

$$f^f_{od}(P) \geq 0 \qquad\qquad \forall P \in \mathcal{P}^1_{od,t}, \forall od \quad \text{(B13c)}$$

where

$$x_{e,t} = \sum_{od} \sum_{P \ni e \cap P \in \mathcal{P}^1_{od,t}} f^f_{od}(P) \qquad\qquad \text{(B13d)}$$

where $f^f_{od}(P)$ is the function to estimate the flow between origin $o$ and destination $d$ on path $P$. While $\mathcal{P}^1_{od,t}$ refers to the set of $od$-paths where some flow is still possible, $\mathcal{P}^0_{od,t}$ refers to the set of $od$-paths where no flow is possible. The demand constraints Eq. (B13b) state that the flow on a given $od$-pair has to equal the demand $d_{od} \geq 0$, for all $od$. The non-negativity constraints Eq. (B13c) are required to ensure that the solution of the program will be physically meaningful.

In terms of the edge cost function $C^{f,T}$, which estimates the travel time $t^T_{e,t}$ through edge $e$ at time $t$ when using the corresponding traffic flow as an input, has been defined using the formulation proposed by the Bureau of Public Roads (1964):

$$C^{f,T}(x_{e,t}) := (1 - \phi_{e,t}) \cdot t^T_{e,0} \left( 1 + \alpha_e \left( \frac{x_{e,t}}{(1 - \langle\lambda\rangle_{e,t}) \cdot y_{e,0}} \right)^{\beta_e} \right) \qquad\qquad \text{(B14)}$$

where $t^T_{e,0}$ is the initial free flow travel time, $y_{e,0}$ the initial edge capacity, and $\alpha_e$ and $\beta_e$ are parameters for calibration, with typical values $\alpha = 0.15$ and $\beta = 4$.

**B13    Restoration model**

The restoration model was defined as a set of consecutive tasks, executed after the occurrence of the hazard events. The model interacted with the functional loss models and traffic models to properly output the indirect costs over time. Interventions were performed after the network was inspected (1 day) and if the functional loss of exceeded a threshold of 10%, i.e., $\langle\lambda\rangle_{e,t} > 0.1$. The number of work crews available was assumed to be 10.





---

**Algorithm 1:** Restoration

---

**1** **for** $e$ **in** $\mathcal{E}$ **do**

**2** $\quad$ **if** $\langle\lambda\rangle_{e,t} > 0.1$ **then**

**3** $\quad\quad$ label $e_{\text{label}} \leftarrow$ `"need of restoration"`

**4** $\quad$ **else**

**5** $\quad\quad$ label $e_{\text{label}} \leftarrow$ `"restored"`

**6** rank $\mathcal{E}$ given $e_{\text{label}}$, $P^0_{od,t}$, $x_{e,0}$

**7** set $\tau_R \leftarrow 0$

**8** **while** $\forall e_{label} \neq$ *"restored"* **do**

**9** $\quad$ **if** *work crew is available* **then**

**10** $\quad\quad$ assign available work crew to $e \in \mathcal{E}$

**11** $\quad\quad$ set $e_{\text{label}} \leftarrow$ `"under restoration"`

**12** $\quad\quad$ set $t \leftarrow \tau_R + \langle\tau_e\rangle$

**13** $\quad$ evaluate indirect costs $\quad\quad\triangleright$ Equation (6)

**15** $\quad$ **if** $t = \tau_R$ **then**

**16** $\quad\quad$ label $e_{\text{label}} \leftarrow$ `"restored"`

**17** $\quad\quad$ set $\lambda_e \leftarrow 0$

**18** $\quad$ $\tau_R + 1$

---





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
