# Peer review of "Estimating network related risks: A methodology and an application in the transport sector"

_Natural Hazards and Earth System Sciences, 2017_

## Referee Comment (RC1) · Anonymous Referee #1 · 10 Apr 2018

1. Does the paper address relevant scientific and/or technical questions within the scope of NHESS?

The paper addresses the question of risk related to networks exposed to natural phenomena which is an important issue. However, the paper focuses on application and combination of several physical models in relation with network modelling without really explaining why those models are used and what are the asumptions. The question of models relevance, key issues in modelling are not addressed (why one model in comparison with others existing ones).

2. Does the paper present new data and/or novel concepts, ideas, tools, methods or

results?

Despite of the interest of the research question, the inputs of the approach are not totally clearly described. This could be improved. What is new ? why ?

3. Are these up to international standards?

An extended bibliography has been done indeed. Some references about networks analysis, socio-economic features should be included (. Different criteria can be considered to assess vulnerability, multicriteria decision making methods may be an alternative: this has not been addressed at all. A past intereg project called Paramount has, between others, addressed this kind of issues...

4. Are the scientific methods and assumptions valid and outlined clearly?

The purpose of the paper is risk assessment. The difference between hazard, damage assessment is not clearly described. Main issues are: 1) The description of the methodology is finally unclear. A major reconfiguration of the paper should be done: a classical paper structure (intro, state of art in each domain, gaps , developments, results, discussion) would be better. 2) A global chart showing all methods and connections would be welcome : the paper is difficult to read 3) Many symbols are used all over the text, sometimes not clearly defined or with confusing notation ( e.g. E for event instead of Expectancy...tricky when also dealing with probability). Not all notations defined are used in the text, results etc...Is it useful in that case? If a model is described, we expect to know which data have been put inside, which asumptions are done 4) Several models have been used : description are given in appendix but it is very difficult to understand what where the assumptions and data used. Some models are perhaps not the right ones to model the phenomenon addressed (e.g. scouring modelling requires to use hydraulics models considering solid transport) 5) The approach on networks is finally (apparently) quite simple and based only on population gradient. Many others socio-economic factors (industry, rescue, education access etc...) are of higher interest to assess indirect risks on networks. Those aspects should be

considered: on the contrary, explain why not developed and speak about limits of the approach.

5. Are the results sufficient to support the interpretations and the conclusions?

Several models are used and combined. Links between them, the way they are used, data are not fully described. Figures are not completely clear and supporting the demonstration (e.g. figure5 aggregated simulation, what is aggregation? How is it done?

6. Does the author reach substantial conclusions?

The conclusion claims that yes but it is not completely convincing. Costs are presented as societal effects. One main output would be that it gathers different phenomena but the way it's done remains not clear : are all events equivalent, is there not a question/issue of importance, relevance? How are the events identified, compared one with another?

7. Is the description of the data used, the methods used, the experiments and calculations made, and the results obtained sufficiently complete and accurate to allow their reproduction by fellow scientists (traceability of results)?

No, it may be impossible to reproduce calculations since basic hypothesis of models used are not described. This is one important suggestion that could be done to better explain and understand the process and the added value of using and combining models.

8.Does the title clearly and unambiguously reflect the contents of the paper?

Redundancy in title (networks): should be changed.

9. Does the abstract provide a concise, complete and unambiguous summary of the work done and the results obtained

The integration should be a major objective but the way all methods are combined is

not clear. The demonstration of the usefulness of the approach is not proved since no comparison with classical approaches is done. Why is it better? How does it help decisions?

10. Are the title and the abstract pertinent, and easy to understand to a wide and diversified audience?

Expectations about risk are high. The focus seems to be more on phenomena and hazards.

11. Are mathematical formulae, symbols, abbreviations and units correctly defined and used? If the formulae, symbols or abbreviations are numerous, are there tables or appendixes listing them?

No, a glossary is given (good) but not all symbols are used. Not useful in that case, some of them are not easily understandable. The reader would have to go to initial bibliography. Data which are used should be described.

12.Is the size, quality and readability of each figure adequate to the type and quantity of data presented? Some figures are difficult to read /interpret (e.g. fig 6) = a set of curves. Think to white and black printing. . .

13. Does the author give proper credit to previous and/or related work, and does he/she indicate clearly his/her own contribution? This paper is a result of aFP7 research project with existing published papers with the same authors). The difference and added value description should be improved

14. Are the number and quality of the references appropriate? Many references but some on the key aspect of indirect vulnerability assessment are missing.

A commented version of the paper (hand-written comments) has been done on paper and can be sent to the authors through editor if wanted.

Please also note the supplement to this comment:

https://www.nat-hazards-earth-syst-sci-discuss.net/nhess-2017-446/nhess-2017-446-RC1-supplement.pdf

[Figure]

**Supplement:**

*1) The methodology is unclear. A major reconfiguration of the article has to be done. Several models have been used but the logical connection between them is not obvious...*

*Redundancy*

*2) The approach about networks is finally very simple and based only on population dependent which does not report socio-economic complexity. Any comment about this? — effect of phenomenon not of a hazard.*

**Estimating the risk related to networks: a methodology and an application on a road network**

Jürgen Hackl[1], Juan Carlos Lam[1], Magnus Heitzler[2], Bryan T. Adey[1], and Lorenz Hurni[2]

[1]Institute of Construction and Infrastructure Management, ETH Zurich, 8092 Zurich, Switzerland
[2]Institute of Cartography and Geoinformation, ETH Zurich, 8092 Zurich, Switzerland
*Correspondence to:* Jürgen Hackl (hackl@ibi.baug.ethz.ch)

**Abstract.** Networks, such as transportation, water, and power, are critical lifelines to society. Managers plan and execute interventions to guarantee the operational state of their networks under various circumstances, including after the occurrence of (natural) hazard events. Creating an intervention program demands knowing the probable consequences (i.e., risk) of the various hazard events that could occur to be able to mitigate their effects. This paper introduces a methodology to support network managers in the quantification of the risk related to their networks. The method emphasizes the integration of the spatial and temporal attributes of the events that need to be modeled to estimate the risk. This work then demonstrates the usefulness of the methodology through its application to design and implement a risk assessment to estimate the potential impact of flood and mudflow events on a road network located in Switzerland. The example includes the modeling of (i) multiple hazard events, (ii) their physical and functional effects throughout the road network, (iii) the functional interrelationships of the affected objects in the network, (iv) the resulting probable consequences in terms of expected costs of restoration, cost of traffic changes, and duration of network disruption, and (v) the restoration of the network.

*3) A global chart showing all methods and connection is missing*

*4) Symbols etc... are either missing or explanation disseminated all over the text.*

**1 Context**

Managers of networks, such as transportation, water, and power, have the continuous task to plan and execute interventions to guarantee the operational state of their networks. This also applies in the aftermath of (natural) hazard events. Since the resources available to managers to protect their networks are limited, it is essential for managers to be aware of the probable consequences (i.e., risk) in order to set priorities and be resource-efficient (Eidsvig et al., 2016). Conducting a risk assessment can help identify the probable hazard events, and evaluate their impact on networks and their users.

Nonetheless, conducting a risk assessment can be a particularly challenging task due to the large number of *scenarios* (i.e., chains of interrelated events) that need to be taken into account, and their associated probabilities and estimated consequences. Furthermore, multiple types of hazards need to be considered (Komendantova et al., 2014; Mignan et al., 2014; Gallina et al., 2016), along with the complex nature of networks, specifically, their large number of objects, their spatial distribution, and functional interrelationships. Moreover, there are additional spatial and temporal characteristics that need to be considered in building these scenarios, along with methods to model the cascade of events, the network interdependencies, and the propagation of uncertainties (Hackl et al., 2015).

*A classical paper structure [Intro - Stand of Art - Res - Results - Discussion] would be better*

*↳ other references [Paramount project... , ...] [Glaps, ... ]*

*References to network features and vulnerabilities*

*State of the art should be extended →*
*; Why is your approach new? both? →*

[Figure]

*other row*

[revised manuscript text omitted]

*"The measure depend on the problem investigated"... whatelse?*
*not very useful statement.*

*Define relationships*

[Figure]

*[handwritten: infrastructure object event ?]*

| Event | Description | Example |
|-------|-------------|---------|
| Source | An event that may lead to a hazard event. | Rainfall, tectonic plate movement |
| Hazard | An event that may lead to an infrastructure object event, and some times, to another hazard event (e.g., earthquake triggers landslides). | Flood, earthquake, landslide |
| Infrastructure object | An event that is a change in the infrastructure object that may lead to a change in network use or a change in human behavior. | Pier scour, pavement inundation, mudblocking |
| Network | An event that is a change in how the network can be used that may lead to a change in human behavior. | Road closures, reduced speed limits |
| Societal | An event that is a change in human behavior (to which a value can be placed). | direct and indirect costs |

**Table 1.** Basic event types.

*[handwritten: 9!]* *[handwritten: 9!]*
*[handwritten: Name is confusing. Infrastructure is not an event.]*
*[handwritten: The definition of events has to be re-defined.]*

Models representing the relationships between the events help build the scenarios, and estimate their probabilities of occurrence. This sub-task is focused on (i) determining a target level of relationship accuracy, and (ii) developing these relationships using available data. Extra effort should be spent on describing these relationships in more detail when approximations are not adequate as exposed by the obtained results.

5 *Define scenarios*

The definitions of events and relationships provide the basis for constructing scenarios. The identification of the scenarios to be analyzed as part of the risk assessment should be done with care (e.g., without explicit estimation of their probability of occurrence or a consequence value beforehand to avoid a biased selection), knowing that excluding specific scenarios may result in an incorrect estimation of risk.

10 *Determine tools*

Once the scenarios to be analyzed are determined, the specific tools, models and approaches to be used to estimate the risk are identified. This includes the selection of the software packages to be used if computer support is required, or a decision to develop software tailored to the specific needs of the risk assessment. This task ends with the selection of tools, models, and software.

15 **3.2.4 Estimate risk** *[handwritten: of what]* *[handwritten: ?!]*

In this task, the probabilities of scenarios and their consequences are estimated, and aggregated when desired. This task can be undertaken with varying degrees of detail and with or without computer support, depending on the problem, and the data, information, and resources available. This task ends with the estimation of the risk, their uncertainty and their sensitivity to modeling assumptions.

*[handwritten: varying degrees ?!]*

*[handwritten: depending on problem ..]*

*[handwritten: Risk to objects vs. risk to functions ... two different issues ...]*

[revised manuscript text omitted]

*[handwritten annotation: → no representation of the phenomena addressed - which model is able to model Hydraulic scouring ?]*

10 Assuming a negligible contraction of the river bed cross-sections (Gehl and D'Ayala, 2015), only local (pier) scour was considered for the example, resulting in the selection of five bridges that could fail as a result of this phenomenon. To qualify different levels of local scour, four damage (limit) states were defined as described in Table 2.

| State | Label | Description |
|-------|-------|-------------|
| 0 | *operational* | no changes in bridge response |
| 1 | *monitored* | first noticeable changes in bridge response |
| 2 | *capacity-reduced* | significant changes in the bridge response |
| 3 | *closed* | lack of pier stability to support the bridge |

**Table 2.** Damage states for bridge local scour.

[revised manuscript text omitted]

10    should be considered. Their cost functions $C^{f,dc}$ and $C^{f,ic}$ are associated with the modeled societal events $E^{soc}$ that occur as a result of infrastructure object events $E^{inf}$ and network events $E^{net}$, and can be traced back to hazard events $E^{haz}$ and source events $E^{src}$. This representation can also be used when considering multiple hazard events.

$$\mathcal{R} = \int_{E^{haz}} \mathbb{P}[E^{haz}|E^{src}] \cdot (C^{f,dc}(E^{soc}|E^{inf}) + C^{f,ic}(E^{soc}|E^{net})) \tag{3}$$

Direct costs are the intervention costs for the affected objects, which are required to return those objects to states where

15    these can function again as originally intended, taking into consideration how each individual object will be restored and the sequence in which the objects will be restored. Indirect costs are those related to loss of connectivity and a deficient level of service, however that is defined for a given network (e.g., temporal prolongation of travel for road networks). Since the spatial and temporal correlation between events are considered, risk can be said to be spatially and temporally distributed (i.e., risk estimates vary in space within a defined area of study, and in time with a defined period of analysis).

20    Furthermore, network managers may be often interested in investigating the effect of hazard loads on their critical objects. Therefore, managers need to be given the option to select hazard events based on the periodicity of the manifested site-specific hazard loads (i.e., managers may not be as concerned with selecting a hazard event based on the return period of the preceding source event). To meet this need, the presented risk equation (3) can be modified by first selecting the hazard events according to the return periods related to the site-specific loads (i.e., the probability of occurrence of an scenario $\mathbb{P}[E^{src}|E^{haz}]$ is determined

25    by source event $E^{src}$ leading to an intended hazard event $E^{haz}$ as defined by Hackl et al. (2017)).

**4.8.2 Direct costs**

Only restoration costs were considered as direct costs. For each section $n$ in a damage state $ds$, a restoration intervention was assigned. Associated with each intervention were (i) the capacity losses due to the execution of the intervention, (ii) the length of time required to execute the intervention $\tau_n \geq 0$, and (iii) the cost of the intervention $c_n \geq 0$. This cost was composed of a

30    fixed part $c_n^{fix}$ (e.g., site setup) and a variable part $c_n^{var}$ (e.g., mu/m$^2$ of pavement, mu/m$^3$ of concrete, where mu stands for

monetary units). Based on the derived fragility functions and the observed time-dependent intensity measures $\Xi_t$, the expected restoration costs $\langle c \rangle_{n,t}$ and durations $\langle \tau \rangle_{n,t}$ for each section were calculated.

$$\langle c \rangle_{n,t} = \mathbb{E}\left(c_n | \Xi_t\right) = \sum_{ds} \mathbb{E}\left(c_n^{\text{fix}} + c_n^{\text{var}} | ds\right) \mathbb{P}[ds | \Xi_t] \tag{4}$$

$$\langle \tau \rangle_{n,t} = \mathbb{E}\left(\tau_n | \Xi_t\right) = \sum_{ds} \mathbb{E}\left(\tau_n | ds\right) \mathbb{P}[ds | \Xi_t] \tag{5}$$

The overall expected direct costs $c^{\text{dc}}$ was the sum of the expected direct costs for each intervention executed. It was assumed that intervention costs are not affected by the selected restoration program.

$$c^{\text{dc}} = \sum_n \max_t (\langle c \rangle_{n,t}) \tag{6}$$

Cost estimates were based on Staubli and Hirt (2005) and from a survey conducted by D'Ayala and Gehl (2015). For each object type and damage state, a restoration strategy was derived, and for each strategy, cost[4] and duration values were approximated (Table 7).

| Event | State | Duration | Fixed costs | Variable costs |
|---|---|---|---|---|
| | | [hour/pier] | [mu] | [mu/pier] |
| bridge | 1 | 60 | 16,000 | 24,000 |
| local | 2 | 135 | 30,000 | 40,000 |
| scour | 3 | 240 | 48,000 | 64,000 |
| | | [hour/m$^2$] | [mu] | [mu/m$^2$] |
| pavement | 1 | 0.005 | 3,500 | 16.50 |
| mudblocking | 2 | 0.009 | 9,600 | 165.00 |
| | 3 | 0.015 | 14,400 | 325.00 |
| | | [hour/m$^2$] | [mu] | [mu/m$^2$] |
| pavement | 1 | 0.005 | 3,500 | 16.50 |
| inundation | 2 | 0.009 | 9,600 | 165.00 |
| | 3 | 0.015 | 14,400 | 325.00 |

**Table 7.** Restoration costs and durations.

[4]Costs taken from the literature are adjusted to 2017 price levels. To avoid overinterpreting the specific values that were in the example, monetary units are used instead of real currency.

[Figure]

**4.8.3 Indirect costs**

The indirect costs were comprised of costs for the temporal prolongation of travel and costs due to a loss of connectivity. The overall indirect costs $c^{\mathrm{ic}}$ were measured as the difference between indirect costs at time $t$ and the indirect costs at time 0 when the network was fully functional.

$$\quad c^{\mathrm{ic}} = \sum_t \left[ \sum_{e \in \mathcal{P}^1_{od,t}} C^{f,\mathrm{pt}}(x_{e,t}) + C^{f,\mathrm{lc}}(\mathcal{P}^0_{od,t}) \right] \qquad (7)$$

where $C^{f,\mathrm{pt}}$ was a cost function dependent on the edge traffic flow $x_{e,t}$ in time $t$ through edge $e$ that was part of the set of feasible paths $\mathcal{P}^1_{od,t}$ identified in time $t$, and $C^{f,\mathrm{lc}}$ was a cost function dependent on a loss of connectivity, which was determined based on the set of unfeasible paths $\mathcal{P}^0_{od,t}$ identified in time $t$.

*Temporal prolongation of travel*

10      The cost function attributed to traffic flow included sub-functions to estimate the costs related to travel time $C^{f,\mathrm{tt}}$ and vehicle operation $C^{f,\mathrm{vo}}$.

$$C^{f,\mathrm{pt}}(x_{e,t}) = C^{f,\mathrm{tt}}(x_{e,t}) + C^{f,\mathrm{vo}}(x_{e,t}) \qquad (8)$$

[revised manuscript text omitted]

The logical frame is not described. One does not see which model is used, in connection with which one etc....?

**B1 Geodata**

To represent geospatial elements, including infrastructure objects and natural phenomena, within a computational environment, an entity-based (vector) approach and a continuous field (raster) approach were used. First, the entity-based approach views space as a place to be populated by entities with clearly defined spatial boundaries and associated properties (e.g., an edge within a road network and its type of use). Second, the continuous field approach typically represents natural phenomena as a set of spatially varying values of some attribute, such as precipitation or elevation.

Various geodata from different sources were necessary to construct and operate the different models implemented. Many of these were extracted from the VECTOR25 dataset (Swisstopo, 2015), the main source for topographic maps of Switzerland with a scale denominator of 25,000. A digital terrain model (DTM) of 16 m × 16 m resolution based on the DTM Amtliche Vermessung was used to incorporate elevation data in this application. All geodata were given or transformed in the Swiss coordinate reference system CH1903/LV03 (EPSG code:21781).

All these models come from existing literature. Their description is short. We do not see the results, their practical implementation. The reader should go in any of the reference to check etc... No critical analysis of those models (limits etc...)

[revised manuscript text omitted]

---

## Referee Comment (RC2) · Anonymous Referee #2 · 1 May 2018

The authors address the topic of network infrastructure exposed to natural hazard risk taking the road network around Chur, Switzerland, as an example, and as such, the topic is of relevance for the target journal. Nevertheless, the manuscript shows some weaknesses which are discussed below.

The manuscript takes a systemic viewpoint by introducing the individual steps of risk analysis for networks (which is not a new task, by the way). Nevertheless, a sound review is missing, which makes it difficult to judge whether or not the authors provide added value to the ongoing discussions of this community (some journals are particularly dedicated to network risk and transportation infrastructure, and it would be good

[Figure]

to see the application of the authors in relation to other approaches coming from the hazard community).

With respect to the hazard type, the authors are not consistent in what process type they model (Debris flows? Landslides? Floods?). In one section, the study seems to be limited to one process type only, while the Figures clearly indicate different hazard types ("The purpose of the assessment itself was to quantify the risk of a complete chain of events over space and time, from source events to their consequences, considering: rainfall, runoff, flooding, mudflows, physical damages and functional losses to bridges and roads (road galleries, tunnels and other structures were not of primary concern), traffic changes and restoration works"). Depending on the different hazard characteristics (floods – spatial extent and duration, debris flows and landslides – local occurrence and interruption, maybe damage to the road infrastructure, mudflows – do they really occur in the Chur region?) it remains unclear what exactly the impact will be. Even if they use respective models (and maybe just take the results as an input for network interruption), this should be clarified. The same holds for the determination of direct and indirect effects. It would be better to restrict this study on a clearly named sample of effects so that the potential readers can easily follow the concept.

With respect to the traffic flow model it remains unclear what exactly the input data is. Moreover, I cannot accept a contributions stating that "The methodology is described in detail in Hackl et al. (2016)" simply because the potential reader should get an overview by reading through the current contributions. So at least the main steps have to be summarized here. As such the entire manuscript reads more like a report than a scientific paper – also because of the repetitions that are presented in multiple sections (Floods are a challenge throughout Europe..., just to provide one example).

Irrespectively of the models used, it would be good to only focus on the network reliability (because a blockage can be of any origin, even road closure due to the WEF in Davos, to give another example).

The title of this contribution suggests that the focus is on network reliability, not on the modelling (and more or less proper representation) of hazards affecting the road network This would also sole the issues addressed in the first part ("too many scenarios. . ."). So methodically, the focus should be placed on the following three items: (a) object-level functional losses leading to network functional loss, considering the topology of the network, (b) object-level physical damages and object-level functional losses leading to restoration works, considering specific restoration criteria, and (c) network functional losses leading to changes in traffic flow, supported by an origin-destination matrix.

Sentences such as "Simulation-based risk assessments require the coupling of multiple heterogeneous models, where a given model encapsulates the behavior and state of a part of the system" do not provide added value – if the entire manuscript could be re-structured using a classical scheme these statements can go to the introduction and have to be underpinned by appropriate references.

The underlying traffic model is tricky and should be described in more detail. What exactly is the data used, and what are the assumptions? As far as I know, Chur is an economic hub for the entire region; so how did the assumptions made mirror the real behavior of commuters? This is central since when the authors compute risk, these figures define us the exposure.

So my suggestion is to re-organise the manuscript in the following way: Introduction: sound review of transport network risk to natural hazards, then categorizing the different approaches in network representation and modelling, then presenting the added value based on a identified gap) of the chosen model. Method description of the model and the main assumptions (data used). Results Discussion: limitations, uncertainties, etc.

Additionally, references are outdated, e.g., Eidsvig, U. M. K., Kristensen, K., and Vangelsten, B. V.: Assessing the risk posed by natural hazards to infrastructures, Nat.

Hazards Earth Syst. Sci. Discuss., pp. 1–31, doi:10.5194/nhess-2016-89, 2016 has already been finally published.

Moreover, references to the transport infrastructure are biased towards the contributions of the authors, here we need a broader review and a clear statement of research gaps and needs, and niches to be filled by the current contribution.

English needs a sound proofread by a native speaker.

Given the shortcomings of the current version, I kindly would like to suggest a rejection and encourage the authors to re-submit once the storyline is clear and focused on what has been promised in the title. Alternatively, the authors may wish to choose an alternative journal, presumably from the network modelling community.

---

## Author Comment (AC1) · 12 Jul 2018

Thank you very much for considering our manuscript for possible publication in the Journal of Natural Hazards and Earth System Sciences. We appreciate your careful review of the manuscript and the well structured constructive feedback received. Please find our response below that includes descriptions of the resulting changes in the manuscript. The revised manuscript, the changes made and the supplementary files can be found in the appendix of this commentary[1]. We trust that these changes have improved the work, making it suitable for publication in your journal. We are

[1]https://www.nat-hazards-earth-syst-sci-discuss.net/nhess-2017-446/nhess-2017-446-AC1-supplement.zip

looking forward to your response.

**General response to all reviewers' comments**

Based on the feedback received, we identified three major areas of concern.

1. *The use of an unconventional paper structure*

   The initial idea was to show readers three times the proposed process, each time at a different level of detail: the generic methodology, the application of the methodology, and the modelling. This approach led to the reuse of section headings in some cases and a scattered literature review. We agree with the reviewers that this presentation can cause confusion. Therefore, to improve its readability, we restructured our manuscript as follows:

   (a) (1. Introduction) – We introduced the problem, and included a cohesive literature review of works focused on estimating transportation network related risk due to natural hazards. Additionally, we highlighted the open research in this field, and clearly pointed out our main contributions.

   (b) (2. Methodology) – We reduced the generic methodology to a minimum, and linked this part to the upcoming example, showing the risk assessment in a single instance. Also, we introduced in this section the most important definitions used in the manuscript (e.g., risk, consequences, events) to facilitate the understanding of the content.

   (c) (3. Application) – We removed all redundant content related to the methodology since a connection between the methodology and the example had already been established in the Section 2 (see point (b)). Furthermore, based on the methodology overview in Section 2 (new Figure 1), we explained the modules and models with greater consistency (i.e., why a model is necessary, how a model depends on other models, which model outputs are the most important). We removed all unnecessary mathematical definitions

from the manuscript since (i) these do not contribute to the understanding of the content and (ii) such definitions make the manuscript very hard to follow. For reasons of completeness, however, we attached the mathematical definitions as supplementary files to the paper (see point (h)).

(d) (4. Results) – We restructured this section by subdividing it into three parts: (i) results of a single scenario, (ii) results of multiple scenarios with the same return period, (iii) and results of multiple scenarios with multiple return periods.

(e) (5. Discussion) – We kept the general discussion about the methodology and the models used.

(f) (6. Conclusions) – We made minor additions to this section to expand on possible future work.

(g) (A. Appendix) – As mentioned in Section 3 (see point (c)), we moved the mathematical definitions to supplementary files for interested readers. The remaining content (mainly tables) was moved to an adequate location in the main text.

(h) (S. Supplementary files) – The mathematical definitions were put into supplementary files, which required adding information on the inputs needed from other models, the outputs produced for other models and the model specific requirements (i.e., which data and data format is needed to run the models; e.g., DTM, land-use data). Additionally, we added multiple images to the supplementary files, which could not be included in the result section (e.g., all time steps of the spatial-temporal system evolution).

2. *Lack of clear focus*

In the original manuscript, we were ambitious by including a large amount of content into the paper. This, combined with the unconventional structure of the

manuscript, impaired the clear message of the manuscript. To address this, we made the following changes:

(a) In the introduction section, we clearly stated the four main contributions of our work: (i) the modelling of a complete chain of events, from a source event to its corresponding societal event, including the link between natural hazards, transportation networks and society, (ii) the estimation of socio-economic impacts (indirect consequences) due to changes in the traffic flow, (iii) the consideration of uncertainties through a simulation-based approach, and (iv) the introduction of a novel simulation engine, which allows the easy coupling of different models.

(b) The content related to the methodology was linked to the application and the use of the simulation engine. Any redundant and unnecessary content was removed.

(c) The application section was "cleaned-up". Some of the models in the initial manuscript were supported by equations and variables (e.g., direct and indirect costs), which were not necessary, and undermined the value of the simulation engine (i.e., models can be swapped as long as input and output remain the same). Therefore, we removed the all unnecessary mathematical content.

3. *Understated link between natural hazards and network risks*

In the initial manuscript, we did not discuss this point in fine detail. To highlight this link, we added additional paragraphs to our literature review, addressing the question of network vulnerability (in general and in the context of natural hazards) and pointing out the advantages and disadvantages of current methods. We also added several examples that explained why multi-hazard spatial-temporal models need both, hazard and network models.

Since we made major changes to the manuscript, in some instances, we will address the reviewers' comments by referencing our general response above.

**Reviewer 1's comments**

*Main remarks*

1. *Does the paper address relevant scientific and/or technical questions within the scope of NHESS?*

   The paper addresses the question of risk related to networks exposed to natural phenomena which is an important issue. However, the paper focuses on application and combination of several physical models in relation with network modelling without really explaining why those models are used and what are the asumptions. The question of models relevance, key issues in modelling are not addressed (why one model in comparison with others existing ones).

   Indeed, the initial manuscript lacked clear statements on why the models used were chosen, what their input and output were, and what the underlying assumptions were. At the same time, we missed to give a proper introduction on the simulation engine, which allows to couple different models regardless of their internal structures, as long as the input and output needed to relate them are defined.

   With the new structure of the paper, we focused on the events and their relationships, and not on the models used since these can be swapped easily (e.g., the in-house made 1d-flood model can be replaced by a 2d Basement model, the in-house made traffic model can be replaced by a VISUM model or MATSim model). Due to computational and legal reasons and for further studies on uncertainty quantification, we prioritized the use of in-house models.

   Having said this, we added a restructured description of our models to the sup-
plementary files. There, we added information on the inputs needed from other models, the outputs produced for other models, and the model specific requirements (i.e., which data and data format are needed to run the models; e.g., DTM, land-use data), and their underlying assumptions.

2. *Does the paper present new data and/or novel concepts, ideas, tools, methods or results?*

Despite of the interest of the research question, the inputs of the approach are not totally clearly described. This could be improved. What is new ? why ?

As mentioned in the general section, we dedicated several paragraphs to explaining the research gap and the contribution of our work.

3. *Are these up to international standards?*

An extended bibliography has been done indeed. Some references about networks analysis, socio-economic features should be included (. Different criteria can be considered to assess vulnerability, multicriteria decision making methods may be an alternative: this has not been addressed at all. A past intereg project called Paramount has, between others, addressed this kind of issues...

Thank you for this comment. In the initial manuscript, we missed this opportunity. Now, we extended our literature review with other common concepts of network vulnerability and positioned our work accordingly. Unfortunately, we could not find recent publications of the Paramount project online. In this regard, we would be pleased if the reviewer could provide us with additional information on the location of those publications or the publications themselves, should their review be still required considering the new literature review.

4. *Are the scientific methods and assumptions valid and outlined clearly?*

The purpose of the paper is risk assessment. The difference between hazard, damage assessment is not clearly described. Main issues are: 1) The description of the methodology is finally unclear. A major reconfiguration of the paper should be done: a classical paper structure (intro, state of art in each domain, gaps , developments, results, discussion) would be better. 2) A global chart showing all methods and connections would be welcome : the paper is difficult to read 3) Many symbols are used all over the text, sometimes not clearly defined or with confusing notation ( e.g. E for event instead of Expectancy. . .tricky when also dealing with probability). Not all notations defined are used in the text, results etc. . .Is it useful in that case? If a model is described, we expect to know which data have been put inside, which asumptions are done 4) Several models have been used : description are given in appendix but it is very difficult to understand what where the assumptions and data used. Some models are perhaps not the right ones to model the phenomenon addressed (e.g. scouring modelling requires to use hydraulics models considering solid transport) 5) The approach on networks is finally (apparently) quite simple and based only on population gradient. Many others socio-economic factors (industry, rescue, education access etc. . .) are of higher interest to assess indirect risks on networks. Those aspects should be considered: on the contrary, explain why not developed and speak about limits of the approach.

Thank you for this valuable feedback, we addressed all your points:

(1) As mentioned in the general response, we changed the paper structure to a classical form (i.e., 1. Introduction with literature review, problem statement and contribution, 2. Methodology, 3. Application, 4. Results, 5. Discussion and 6. Conclusions).

(2) We replaced the representation of the generic risk assessment methodology (former Figure 1) with an overview of all models and their connections as suggested (new Figure 1).

(3) Indeed, too many symbols were used with most of them only once. We removed all model specific variables and equations to the supplementary files, and only kept the most important variables and equations, which were directly explained in the text so that the reader does no longer have to switch forwards (to the nomenclature) and backwards (to the section).

(4) We restructured the way models are described and gave more insights on the inputs, output and assumptions used.

(5) This is true. Currently, only indirect costs associated with temporal prolongation of travel and loss of connectivity are quantified. We are working on the extension to model and quantify other socio-economic factors, as mentioned above (e.g., industry, rescue, education access). We added this point to our discussion section and as an outlook for further research.

5. *Are the results sufficient to support the interpretations and the conclusions?*

Several models are used and combined. Links between them, the way they are used, data are not fully described. Figures are not completely clear and supporting the demonstration (e.g. figure5 aggregated simulation, what is aggregation? How is it done?

For the first point please see above. We added more details about the models and also visualized their interdependencies in Figure 1.

We also agree with the second point: some of the figures were hard to understand without additional information. Therefore, we made some changes to the figures:

[Figure]

(a) Figure 1. was replaced with an schematic overview of the investigated chain of events, showing the models and their relationships.

(b) A new Figure 3. was added, illustrating the terminology used to describe network, objects, sections and subsections used in the manuscript.

(c) Former Figure 3. was split up into two separate figures, showing the fragility curves (Figure 4.) and the functional loss curves (Figure 5.).

(d) An additional figure (Figure 6.) was added, showing the use of functional loss functions during the process of deriving a damaged network from damaged infrastructure sections.

(e) Former Figure 4. (now Figure 7.) was replaced by a figure showing the considered events in two time steps rather than eighteen. The evolution of the events considering all eighteen time steps was added to the supplementary files.

(f) The content of former Figure 5. (now Figure 8.) was extended to include the aggregated direct costs.

(g) Former Figure 6. was replaced by a new figure (Figure 9.), showing an analysis of the direct and indirect costs for multiple events of 500-year return period.

(h) Former Figures 7a, 7b and 7c were combined in a single figure (Figure 10a.). In addition to the risk curves, box plots of estimated annualized risk values were added (Figure 10b.).

(i) Former Figure 8 was removed.

Descriptive captions were added to all figures, such that figures can be interpreted without having to reference the main text.

6. *Does the author reach substantial conclusions?*

The conclusion claims that yes but it is not completely convincing. Costs are presented as societal effects. One main output would be that it gathers different phenomena but the way it's done remains not clear : are all events equivalent, is there not a question/issue of importance, relevance? How are the events identified, compared one with another?

«*Costs are presented as societal effects.*» This assumption was used since the traffic costs absorbed during the restoration period could not be directly attributed to a single object. The network manager has multiple options on when and how objects are repaired. Such decisions influence the traffic flow, and therefore, the indirect costs carried by the road users.

In this paper, we assumed that direct and indirect costs were equally important since the major task was to quantify the risks. In reality, decision makers such as network managers and politicians assign different importance to these costs.

For illustration purposes, we chose restoration costs to illustrate direct consequences and temporal prolongation of travel and loss of connectivity to illustrate indirect consequences. To compare these consequences we assigned (unweighted) monetary values. In the future, we plan to incorporate additional measures such as business interruption, and accessibility to hospitals, among others.

7. *Is the description of the data used, the methods used, the experiments and calculations made, and the results obtained sufficiently complete and accurate to allow their reproduction by fellow scientists (traceability of results)?*

No, it may be impossible to reproduce calculations since basic hypothesis of models used are not described. This is one important suggestion that could be done

to better explain and understand the process and the added value of using and combining models.

In the new version of the manuscript, we gave more information about the models used. This included the inputs and outputs, the data requirement of the models (e.g. DTM, land-use data) and some of the underlying assumptions.

In the future, after proper testing, documentation, refactoring and clean up of the code base, we would like to release the source code of our simulation engine.

8. *Does the title clearly and unambiguously reflect the contents of the paper?*

Redundancy in title (networks): should be changed.

Thank you for this comment. We changed our title to "Estimating network related risks: A methodology and an application in the transport sector", and removed the second reference to "network".

9. *Does the abstract provide a concise, complete and unambiguous summary of the work done and the results obtained*

The integration should be a major objective but the way all methods are combined is not clear. The demonstration of the usefulness of the approach is not proved since no comparison with classical approaches is done. Why is it better? How does it help decisions?

In Section 2. (Methodology) we refocused the content on the implementation of the methodology. In combination with Section 3. (Application), the reader should now have a clearer picture on how all elements of the methodology fit together.

To avoid making article longer, we omitted a quantitative comparison with other
classical approaches. We added, however, additional literature about the state-of-the-art network (vulnerability) analysis. A comparison with other approaches should be part of further research, and is stated as such in the manuscript.

10. *Are the title and the abstract pertinent, and easy to understand to a wide and diversified audience?*

Expectations about risk are high. The focus seems to be more on phenomena and hazards.

Please refer to our general response. We hope that the restructured manuscript and the tighter focus on the coupling of the different events, their relationships, and their effects on networks address this concern.

11. *Are mathematical formulae, symbols, abbreviations and units correctly defined and used? If the formulae, symbols or abbreviations are numerous, are there tables or appendixes listing them?*

No, a glossary is given (good) but not all symbols are used. Not useful in that case, some of them are not easily understandable. The reader would have to go to initial bibliography. Data which are used should be described.

We agree with this observation. The initial manuscript included too much (mathematical) information, which was not needed to understand the content. In contrast, this level of information made following the article more difficult. To overcome this issue, we moved most of the equations and variables from the main text into the supplementary files. There, we restructured the information about each model for readers to immediately see where the variables are used and what is behind them (e.g., is it a raster cell value associated with a variable, or is it a function?)

12. *Is the size, quality and readability of each figure adequate to the type and quantity of data presented?*

Some figures are difficult to read /interpret (e.g. fig 6) = a set of curves. Think to white and black printing...

Please refer to Point 5. We made major modifications to the figures in the manuscript.

13. *Does the author give proper credit to previous and/or related work, and does he/she indicate clearly his/her own contribution?*

This paper is a result of aFP7 research project with existing published papers with the same authors). The difference and added value description should be improved

As mentioned above in the general response, we highlighted the main contributions of this work in the revised manuscript.

14. *Are the number and quality of the references appropriate?*

Many references but some on the key aspect of indirect vulnerability assessment are missing.

We added a paragraph to our literature review where we discuss state-of-the-art network vulnerability assessments and how we use the term vulnerability in our work. In contrast to other authors, we consider vulnerability as an inherent attribute of any system and an essential part of the risk assessment. In our manuscript, we made a distinction between costs and damages, and hence, we used the familiar concept of fragility instead of vulnerability.

A commented version of the paper (hand-written comments) has been done on paper and can be sent to the authors through editor if wanted.

Thank you very much for your effort. We incorporated as many comments as possible. If a detailed response to these comments is still needed, please let us know.

The authors would like to thank the editor and the reviewers in advance for their careful evaluation of our manuscript.

On behalf of our research team

Jürgen Hackl
(corresponding author)

Please also note the supplement to this comment:
https://www.nat-hazards-earth-syst-sci-discuss.net/nhess-2017-446/nhess-2017-446-AC1-supplement.zip

---

## Author Comment (AC2) · 12 Jul 2018

Thank you very much for considering our manuscript for possible publication in the Journal of Natural Hazards and Earth System Sciences. We appreciate your careful review of the manuscript and the well structured constructive feedback received. Please find our response below that includes descriptions of the resulting changes in the manuscript. The revised manuscript, the changes made and the supplementary files can be found in the appendix of this commentary[1]. We trust that these changes have improved the work, making it suitable for publication in your journal. We are
* * *
[1]https://www.nat-hazards-earth-syst-sci-discuss.net/nhess-2017-446/nhess-2017-446-AC2-supplement.zip

looking forward to your response.

**General response to all reviewers' comments**

Based on the feedback received, we identified three major areas of concern.

1. *The use of an unconventional paper structure*

   The initial idea was to show readers three times the proposed process, each time at a different level of detail: the generic methodology, the application of the methodology, and the modelling. This approach led to the reuse of section headings in some cases and a scattered literature review. We agree with the reviewers that this presentation can cause confusion. Therefore, to improve its readability, we restructured our manuscript as follows:

   (a) (1. Introduction) – We introduced the problem, and included a cohesive literature review of works focused on estimating transportation network related risk due to natural hazards. Additionally, we highlighted the open research in this field, and clearly pointed out our main contributions.

   (b) (2. Methodology) – We reduced the generic methodology to a minimum, and linked this part to the upcoming example, showing the risk assessment in a single instance. Also, we introduced in this section the most important definitions used in the manuscript (e.g., risk, consequences, events) to facilitate the understanding of the content.

   (c) (3. Application) – We removed all redundant content related to the methodology since a connection between the methodology and the example had already been established in the Section 2 (see point (b)). Furthermore, based on the methodology overview in Section 2 (new Figure 1), we explained the modules and models with greater consistency (i.e., why a model is necessary, how a model depends on other models, which model outputs are the most important). We removed all unnecessary mathematical definitions

from the manuscript since (i) these do not contribute to the understanding of the content and (ii) such definitions make the manuscript very hard to follow. For reasons of completeness, however, we attached the mathematical definitions as supplementary files to the paper (see point (h)).

(d) (4. Results) – We restructured this section by subdividing it into three parts: (i) results of a single scenario, (ii) results of multiple scenarios with the same return period, (iii) and results of multiple scenarios with multiple return periods.

(e) (5. Discussion) – We kept the general discussion about the methodology and the models used.

(f) (6. Conclusions) – We made minor additions to this section to expand on possible future work.

(g) (A. Appendix) – As mentioned in Section 3 (see point (c)), we moved the mathematical definitions to supplementary files for interested readers. The remaining content (mainly tables) was moved to an adequate location in the main text.

(h) (S. Supplementary files) – The mathematical definitions were put into supplementary files, which required adding information on the inputs needed from other models, the outputs produced for other models and the model specific requirements (i.e., which data and data format is needed to run the models; e.g., DTM, land-use data). Additionally, we added multiple images to the supplementary files, which could not be included in the result section (e.g., all time steps of the spatial-temporal system evolution).

2. *Lack of clear focus*

In the original manuscript, we were ambitious by including a large amount of content into the paper. This, combined with the unconventional structure of the

manuscript, impaired the clear message of the manuscript. To address this, we made the following changes:

(a) In the introduction section, we clearly stated the four main contributions of our work: (i) the modelling of a complete chain of events, from a source event to its corresponding societal event, including the link between natural hazards, transportation networks and society, (ii) the estimation of socio-economic impacts (indirect consequences) due to changes in the traffic flow, (iii) the consideration of uncertainties through a simulation-based approach, and (iv) the introduction of a novel simulation engine, which allows the easy coupling of different models.

(b) The content related to the methodology was linked to the application and the use of the simulation engine. Any redundant and unnecessary content was removed.

(c) The application section was "cleaned-up". Some of the models in the initial manuscript were supported by equations and variables (e.g., direct and in-direct costs), which were not necessary, and undermined the value of the simulation engine (i.e., models can be swapped as long as input and output remain the same). Therefore, we removed the all unnecessary mathematical content.

3. *Understated link between natural hazards and network risks*

In the initial manuscript, we did not discuss this point in fine detail. To highlight this link, we added additional paragraphs to our literature review, addressing the question of network vulnerability (in general and in the context of natural hazards) and pointing out the advantages and disadvantages of current methods. We also added several examples that explained why multi-hazard spatial-temporal models need both, hazard and network models.

Since we made major changes to the manuscript, in some instances, we will address the reviewers' comments by referencing our general response above.

**Reviewer 2's comments**

The authors address the topic of network infrastructure exposed to natural hazard risk taking the road network around Chur, Switzerland, as an example, and as such, the topic is of relevance for the target journal. Nevertheless, the manuscript shows some weaknesses which are discussed below.

Thank you very much. Please find below a summary of changes made to strengthen our paper in response to your feedback.

*Main remarks*

1. The manuscript takes a systemic viewpoint by introducing the individual steps of risk analysis for networks (which is not a new task, by the way). Nevertheless, a sound review is missing, which makes it difficult to judge whether or not the authors provide added value to the ongoing discussions of this community (some journals are particularly dedicated to network risk and transportation infrastructure, and it would be good to see the application of the authors in relation to other approaches coming from the hazard community).

   Please refer to the general response. In the initial manuscript, we missed the opportunity to give a detailed discussion about the link between natural hazards and network risks. To overcome this issue, we added additional paragraphs to our literature review, addressing the question of network vulnerability (in general and in context of natural hazards) and pointing out the advantages and disadvantages of current methods. We also added several examples, explaining why multi-hazard spatial-temporal models are in need of network models in addition to hazard models.

2. With respect to the hazard type, the authors are not consistent in what process

type they model (Debris flows? Landslides? Floods?). In one section, the study seems to be limited to one process type only, while the Figures clearly indicate different hazard types ("The purpose of the assessment itself was to quantify the risk of a complete chain of events over space and time, from source events to their consequences, considering: rainfall, runoff, flooding, mudflows, physical damages and functional losses to bridges and roads (road galleries, tunnels and other structures were not of primary concern), traffic changes and restoration works"). Depending on the different hazard characteristics (floods - spatial extent and duration, debris flows and landslides - local occurrence and interruption, maybe damage to the road infrastructure, mudflows - do they really occur in the Chur region?) it remains unclear what exactly the impact will be. Even if they use respective models (and maybe just take the results as an input for network interruption), this should be clarified. The same holds for the determination of direct and indirect effects. It would be better to restrict this study on a clearly named sample of effects so that the potential readers can easily follow the concept.

In our study, we used a multi-hazard approach considering floods and mudflows, both triggered by heavy rainfall. Both are spatio-temporal models, whose variables changed over time, depending on previous time steps. In order to clearly present the relationships between the models, we added a schematic overview of the investigated chain of events (Figure 1.) to the manuscript. Additionally, we restructured the Methodology section (please see general response point 1) to align the methodology with the application, making following the paper easier for readers. Finally, a more detailed description of the models used was added to the supplementary files.

«. . . *do they really occur in the Chur region?*» Unfortunately, such events occur. In the last decades, four major flood events with infrastructure damages over 5 million CHF occurred in this region. This area is also prone to landslides, including debris flows. In fact, the data obtained for the modelling of the mudflow events

is for the area of study. To highlight this, we added a comprehensive description to Figure 2., where some of the past hazard events are illustrated.

We trust that the new structure of the manuscript and the schematic overview of the process (Figure 1.) have improved the readability of the manuscript.

3. With respect to the traffic flow model, it remains unclear what exactly the input data is. Moreover, I cannot accept a contributions stating that "The methodology is described in detail in Hackl et al. (2016)" simply because the potential reader should get an overview by reading through the current contributions. So at least the main steps have to be summarized here. As such the entire manuscript reads more like a report than a scientific paper - also because of the repetitions that are presented in multiple sections (Floods are a challenge throughout Europe. . ., just to provide one example).

Thank you for this valuable comment. This comment prompted us to change the structure of our article (please see general response). As stated there, the initial idea was to present the reader three times the same process, each time at a different level of detail (i.e., generic, application, models). This caused a lot of redundancy, and prevented us from presenting the three processes with the necessary depth. The new manuscript was refocused on the application. We removed unnecessary parts of the generic methodology, and moved unnecessary mathematical definitions to the supplementary files. This allowed us to focus the content and give readers a clear idea of how the proposed approach can be followed.

4. Irrespectively of the models used, it would be good to only focus on the network reliability (because a blockage can be of any origin, even road closure due to the WEF in Davos, to give another example).

Due to the unconventional paper structure and the lack of focus, one major contribution of our work may have been lost in the manuscript. Modelling the complete

disaster chain from a source event to its consequences results in a more in-depth analysis than just considering blockage of any origin. For example, we can quantify the severity of the damages and functional losses, leading to the identification of optimal intervention strategies (e.g., if a mudflow blocked an important road with a couple of hundreds $m^2$, the network manager can estimate how long removing the debris with an excavator would take). Also, since the hazard intensities (e.g., flood inundation depths) change over time, road blockage, which is spatially correlated, also changes over time. To summarize, the consideration of blockages of any origin might be a common approach for network reliability, but more information is needed for network resilience. We addressed this issue in the new Introduction section by adding an additional discussion about commonly used approaches to analyze network vulnerability and reliability, their advantages and drawbacks, as well as the contribution of our work on this front.

5. The title of this contribution suggests that the focus is on network reliability, not on the modelling (and more or less proper representation) of hazards affecting the road network This would also sole the issues addressed in the first part ("too many scenarios..."). So methodically, the focus should be placed on the following three items: (a) object-level functional losses leading to network functional loss, considering the topology of the network, (b) object-level physical damages and object-level functional losses leading to restoration works, considering specific restoration criteria, and (c) network functional losses leading to changes in traffic flow, supported by an origin-destination matrix.

As suggested, we placed higher emphasis in the use of fragility and functional loss functions. Therefore, we created dedicated sections explaining the general idea of both functions and how these are used in the application. Additionally, in Figure 6., we illustrated how object-level functional losses lead to network functional loss, and in Figure 1., we showed the relationships between object-level physical damages and object-level functional losses leading to restoration works,

and how network functional losses lead to changes in traffic flow.

6. Sentences such as "Simulation-based risk assessments require the coupling of multiple heterogeneous models, where a given model encapsulates the behavior and state of a part of the system" do not provide added value - if the entire manuscript could be re-structured using a classical scheme these statements can go to the introduction and have to be underpinned by appropriate references.

   As stated in the general response, we restructured the manuscript and moved such statements to the introduction section.

7. The underlying traffic model is tricky and should be described in more detail. What exactly is the data used, and what are the assumptions? As far as I know, Chur is an economic hub for the entire region; so how did the assumptions made mirror the real behavior of commuters? This is central since when the authors compute risk, these figures define us the exposure.

   We agree that traffic modelling is a science in itself. For the sake of illustration, computational costs, and legal issues, we used an in-house solution: a static user equilibrium traffic assignment model, based on the BPR functions to simulate traffic flow conditions. Although this model is mathematically simple, computationally inexpensive and widely used in literature, it has limitations. The model assumes that travelers have full knowledge of the traffic conditions, which is not the case. It does not account for travel pattern changes after a disruptive event either, although studies show that this behaviour is considerably different than before a disruptive event. However, with the proposed modular framework of the simulation engine, such a model can easily be replaced by a VISUM or MATSim model. In further studies, we plan to replace this simplified traffic model with an agent-based model in order to also quantify other socio-economic factors such as business interruptions, rescue missions, fatalities, and education access, among other relevant factors. For this paper, we added additional information on

the used traffic model to the supplementary files.

8. So my suggestion is to re-organise the manuscript in the following way: Introduction: sound review of transport network risk to natural hazards, then categorizing the different approaches in network representation and modelling, then presenting the added value based on a identified gap) of the chosen model. Method description of the model and the main assumptions (data used). Results Discussion: limitations, uncertainties, etc.

As suggested we changed the structure of our manuscript. For details please have a look at the general response section.

9. Additionally, references are outdated, e.g., Eidsvig, U. M. K., Kristensen, K., and Vangelsten, B. V.: Assessing the risk posed by natural hazards to infrastructures, Nat. Hazards Earth Syst. Sci. Discuss., pp. 1-31, doi:10.5194/nhess-2016-89, 2016 has already been finally published.

Thank you, we updated the references.

10. Moreover, references to the transport infrastructure are biased towards the contributions of the authors, here we need a broader review and a clear statement of research gaps and needs, and niches to be filled by the current contribution.

As part of the restructuring, we have added a section with open research in the field and our contributions to the introduction of the manuscript (see general response 1.a).

11. English needs a sound proofread by a native speaker.

This was done.

12. Given the shortcomings of the current version, I kindly would like to suggest a rejection and encourage the authors to re-submit once the storyline is clear and focused on what has been promised in the title. Alternatively, the authors may

wish to choose an alternative journal, presumably from the network modelling community.

We trust that the revised manuscript is now suitable for publication in NHESS.

The authors would like to thank the editor and the reviewers in advance for their careful evaluation of our manuscript.

On behalf of our research team

Jürgen Hackl
(corresponding author)

Please also note the supplement to this comment:
https://www.nat-hazards-earth-syst-sci-discuss.net/nhess-2017-446/nhess-2017-446-AC2-supplement.zip

---

## Author Response (AR2)

Eidgenössische Technische Hochschule Zürich
Swiss Federal Institute of Technology Zurich

**Department of Civil, Environmental and Geomatic Engineering**
**Institute of Construction and Infrastructure Management**

ETH Zurich
Jürgen Hackl
PhD Candidate
IBI - HIL, Room F 23.1
Stefano-Franscini-Platz 5
8093 Zurich, Switzerland

Phone  +41 44 633 65 10
Fax      +41 44 633 10 88
hackl@ibi.baug.ethz.ch
www.ibi.ethz.ch

Zurich, August 6, 2018

**NHESS-2017-446 - response to the reviewers' comments for the manuscript:**
**Estimating network related risks: A methodology and an application in the transport sector**

Dear Prof. Dr. Fuchs and reviewers,

Thank you very much for accepting our manuscript for publication in the Journal of Natural Hazards and Earth System Sciences. We appreciate your careful review of the manuscript and the well structured constructive feedback received. Please find our response below that includes descriptions of the resulting changes in the manuscript.

**Reviewer 1's comments**

The authors address the topic of network infrastructure exposed to natural hazard risk taking the road network around Chur, Switzerland, as an example, and as such, the topic is of relevance for the target journal.

During their review, the authors clarified the main points I raised during my first review, so from my point of view the paper can be published now.

Thank you very much. Your comments and suggestions have significantly improved the content of our manuscript.

1. I only have some very minor comments: On page 3, line 23 f. the authors state that "with a changing climate, exacerbated by an increase in urbanization, the frequency of extreme hydrometeorological hazard events is expected to rise, impacting economic corridors, disrupting supply chain, and stressing emergency and rescue operations, among other effects." It would be nice to see some references here, in particular because the assumed increase in urbanization is highly variable over space and in time. For CC in the Alps, Keiler et al. (2010) have nicely shown the expected effects, and with respect to the question of urbanization, recent works include those of Fuchs et al. (2017) or earlier works by Fuchs and Bründl (2005) explicitly focusing on the canton of Grisons, Switzerland. The general need of including spatio-temporal dynamics in risk management is also summarized by Fuchs et al. (2013) showing examples from the Swiss Alps.
   Thank you for this comment and the references. We agree, these articles are very interesting and a good addition to our literature review. Therefore we have extended our manuscript with the proposed articles.

Overall, the discussion paper increased in accessibility, and will be a major contribution to the NH community, in particular due to the envisaged multi-hazard approach. So I recommend some minor adjustments as indicated

before final acceptance, and wish the authors good success with their future works.

Thank you very much.

References mentioned as suggestion:

- Fuchs, S., and Bründl, M.: Damage potential and losses resulting from snow avalanches in settlements of the canton of Grisons, Switzerland, Natural Hazards, 34, 53-69, 2005.
- Fuchs, S., Keiler, M., Sokratov, S. A., and Shnyparkov, A.: Spatiotemporal dynamics: the need for an innovative approach in mountain hazard risk management, Natural Hazards, 68, 1217-1241, 2013.
- Fuchs, S., Röthlisberger, V., Thaler, T., Zischg, A., and Keiler, M.: Natural hazard management from a coevolutionary perspective: Exposure and policy response in the European Alps, Annals of the American Association of Geographers, 107, 382-392, 2017.
- Keiler, M., Knight, J., and Harrison, S.: Climate change and geomorphological hazards in the eastern European Alps, Philosophical Transactions of the Royal Society of London. Series A: Mathematical, Physical and Engineering Sciences, 368, 2461-2479, 2010.

—

The authors would like to thank the editor and the reviewers for their careful evaluation of our manuscript.

On behalf of our research team

Jürgen Hackl
(corresponding author)